# TD-Paint: Faster Diffusion Inpainting Through Time-aware Pixel Conditioning

**Tsiry Mayet**
INSA Rouen Normandie, LITIS UR 4108,
F-76000 Rouen, France

**Pourya Shamsolmoali**
University of York, United Kingdom
East China Normal University, China

**Simon Bernard**
Université Rouen Normandie, LITIS UR 4108,
F-76000 Rouen, France

**Eric Granger**
LIVIA, Dept. of Systems Engineering,
ETS Montreal, Canada

**Romain Hérault**
Université Caen Normandie, CNRS, GREYC UMR6072,
F-14000, Caen, France

**Clément Chatelain**
INSA Rouen Normandie, LITIS UR 4108,
F-76000 Rouen, France

## Abstract

Diffusion models have emerged as highly effective techniques for inpainting, however, they remain constrained by slow sampling rates. While recent advances have enhanced generation quality, they have also increased sampling time, thereby limiting scalability in real-world applications. We investigate the generative sampling process of diffusion-based inpainting models and observe that these models make minimal use of the input condition during the initial sampling steps. As a result, the sampling trajectory deviates from the data manifold, requiring complex synchronization mechanisms to realign the generation process. To address this, we propose Time-aware Diffusion Paint (TD-Paint), a novel approach that adapts the diffusion process by modeling variable noise levels at the pixel level. This technique allows the model to efficiently use known pixel values from the start, guiding the generation process toward the target manifold. By embedding this information early in the diffusion process, TD-Paint significantly accelerates sampling without compromising image quality. Unlike conventional diffusion-based inpainting models, which require a dedicated architecture or an expensive generation loop, TD-Paint achieves faster sampling times without architectural modifications. Experimental results across three datasets show that TD-Paint outperforms state-of-the-art diffusion models while maintaining lower complexity. Github code: https://github.com/MaugrimEP/td-paint

## 1 Introduction

Given an image and a binary mask, image inpainting aims to generate the missing region while preserving the semantics of the visible region. This task is challenging because the generated content must not only be coherent with the existing parts of the image but also appear realistic. Additionally, the generation process should be stochastic to produce diverse outputs. An effective inpainting model must also address variations in mask shape and size. Generalizing to unseen masks during training and accurately filling large missing regions further complicates the task.

Diffusion models have shown significant success as generative models (Dhariwal & Nichol, 2021; Rombach et al., 2022), by approximating the distribution of real images through a fixed Markov chain that transforms Gaussian noise into the real image distribution. During training, a forward diffusion process gradually adds noise to an image, and the model is trained to reverse this process, learning to denoise and recover the original image distribution. During generation, the backward diffusion iteratively removes noise from an initial Gaussian noise image. The trained model predicts and removes noise at each step, gradually refining the image until a photorealistic result is achieved.

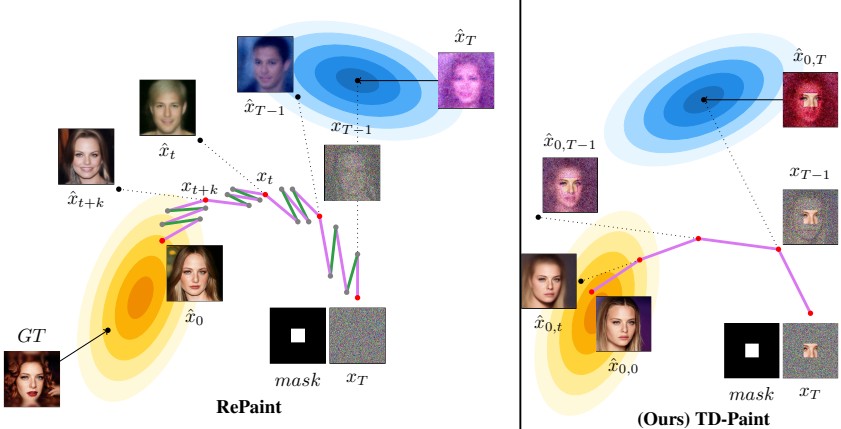

Figure 1: **Comparison of noisy-condition models (e.g., RePaint) and TD-Paint generation processes.** Reverse denoising steps ($p_\theta(x_{t-1}|x_t)$) are depicted in purple lines —, forward noising steps ($q(x_t|x_{t-1})$) are shown in green lines —. Here, $x_t$ represents the input to the diffusion process at step $t$, $mask$ denotes the conditional area (shown in white) and the region to be generated (shown in black), and $\hat{x}_t$ represents the model's prediction at diffusion step $t$. **(left) RePaint (Lugmayr et al., 2022) generation process**. RePaint applies a cycle of reverse and forward diffusion steps. It can be observed that the intermediate steps of the generation process lack consistency, changing from a man with dark hair at $\hat{x}_{T-1}$ to a man with blond hair at $\hat{x}_t$ to a woman at $\hat{x}_{t+k}$. These changes occur due to the synchronization process, where the initially predicted images $\hat{x}_t$ are not well aligned with the given condition. **(right) TD-Paint generation process**. In comparison, TD-Paint can use a clean condition from the beginning of the inpainting process, resulting in a faster and more stable process. Note how the intermediate TD-Paint steps are consistent from one to another.

Methods proposed in the literature have investigated using the standard diffusion model (Lugmayr et al., 2022; Chung et al., 2022) for inpainting by combining the noisy condition with the current generation at each step. This technique has its limitations. While the textures match, it creates disharmony between the conditioning and generation parts. This disharmony comes from the fact that, during the early steps of the generation process, the condition contains a lot of noise that the model cannot leverage. Therefore, the generation moves away from the intended semantics and produces unsatisfactory results. An illustration of such inpainting can be found in Figure 1(left).

To address this limitation, RePaint (Lugmayr et al., 2022) introduced a resampling mechanism that repeats the diffusion steps multiple times, enabling the model to synchronize condition and generation better. Although RePaint produces highly faithful images, this resampling significantly slows the generation process. Indeed, RePaint requires approximately 5k steps to generate a single image, increasing time complexity. An illustration of RePaint's generative process is shown in Figure 1 (left). Other approaches introduce additional constraints at each diffusion step(Chung et al., 2022; Li et al., 2023). For example, (Chung et al., 2022) integrates a correction mechanism that encourages the diffusion path to remain close to the data manifold. This is achieved by minimizing the reconstruction error of the known image region relative to the unknown region.

In contrast to models that use a noisy condition(Lugmayr et al., 2022; Chung et al., 2022), our Time-aware Diffusion Paint (TD-Paint) approach integrates the currently generated sample with a clean condition. Our method uses semantic information from the beginning of the generation process, resulting in a more efficient and cost-effective approach. To achieve this, TD-Paint uses time conditioning derived from the standard formulation of diffusion models. Instead of using a single scalar $t$ for the entire image, TD-Paint assigns a unique $t$ value to each pixel. The known image region is assigned a smaller $t$, indicating lower noise levels. In contrast, the region to be generated is assigned a $t$ value proportional to the current step in the generation process. An illustration of TD-Paint's generative process is provided in Figure 1(right).

**Our main contributions are summarized as follows:**
• We propose a novel noise modeling paradigm for diffusion models that allows for integrating vary-

ing noise levels into the input of diffusion models for the known and unknown regions. TD-Paint exploits time conditioning in diffusion models to achieve faster and higher-quality generation.

• An extensive set of experiments on the challenging CelebA-HQ, ImageNet1K and Places2 datasets demonstrates that our TD-Paint, not only outperforms state-of-the-art diffusion-based models but also surpasses other inpainting methods, including those based on CNNs and transformers. Additionally, the results indicate TD-Paint is a more cost-effective solution for diffusion models.

## 2 RELATED WORKS

Inpainting aims to fill in a missing part of the image realistically. Traditional inpainting methods try to combine techniques to propagate texture and structural information onto the missing parts (Criminisi et al., 2004). Some algorithms (Grossauer, 2004) use large image datasets and assume that the possible semantic space for missing regions is limited. In recent years, deep learning models for inpainting have made impressive progress using two types of generative models, VAEs (Kingma & Welling, 2013) and GANs (Goodfellow et al., 2020; Mirza & Osindero, 2014).

**(a) Single-Stage Inpainting:** Most single-stage methods use the context encoding setting introduced by Pathak et al. (Pathak et al., 2016), with an encoder-decoder setup. A reconstruction loss (L2) ensures global structure consistency, while an adversarial loss ensures the reconstruction is realistic. Global consistency is an important consideration. CNNs are limited by a receptive field that grows slowly with network layers. Many layers are needed for information to travel from one side of the image to the other. Dilated convolution (Yu & Koltun, 2015) has been used by (Iizuka et al., 2017) to increase the receptive field. Partial convolution (Liu et al., 2018) uses mask information to attend only the visible regions. The pyramid context encoder (Zeng et al., 2019) learns an affinity map between regions in a pyramidal fashion. Fourier convolution (Suvorov et al., 2022) aims to provide a global receptive field to both the inpainting network and the loss function. Fast Fourier Convolutions have an image-wide receptive field, which helps with large missing areas. Mask-Aware Transformer (MAT) (Li et al., 2022) is a transformer-based architecture that allows the processing of high-resolution images. A customized transformer block considers only valid tokens, and a style manipulation module updates convolution weights with noise to produce diverse outputs.

**(b) Progressive Image Inpainting:** These methods seek to address global consistency by using coarse-to-fine multi-stage approaches. Multiple generations are possible for a large missing region in single-stage training. Some may have a large pixel-to-pixel distance from the original ground truth, which can be misleading when training models with pixel-wise distance losses. To address this issue, Yun et al. (Yu et al., 2018) proposed a two-stage generative approach. The first stage produces a coarse output optimized with L1 loss incorporating spatial discounting, while the second stage refines the output further using both global and local critics. Gated convolution (Dauphin et al., 2017) has been used in a coarse-to-fine network to learn valid pixels (Yu et al., 2019).

**(c) Prior Knowledge Inpainting:** These methods leverage and mine information from generative models. Prior Guided GAN (PGG) (Suvorov et al., 2022) uses the latent space of a pre-trained GAN and learns to map masked images to this space using an encoder. A masked image can be mapped to a latent code during inference, and the generator can produce a corresponding inpainted image. Deep Generative Prior (Pan et al., 2021) relaxes the frozen generator assumption of GAN inversion methods and proposes progressively refining each layer. PSP (Richardson et al., 2021) uses StyleGAN (Karras et al., 2019) latent space to encode an image into its latent space. Inpainting is formulated as a domain translation task performed in the latent space of StyleGAN, removing adversarial components from the training process.

**(d) Diffusion Model Inpainting:** While GANs have recently shown impressive results, most applications are limited to generating a specific domain. In contrast, Diffusion models have gained traction for image generation; Denoising Diffusion Probabilistic Model (DDPM) (Ho et al., 2020) and Denoising Diffusion Implicit Models (DDIM) (Song et al., 2020a) can generate very diverse and high-quality images. Some unconditional diffusion models have shown the ability to perform zero-shot inpainting (Sohl-Dickstein et al., 2015; Song et al., 2020b) but provide only qualitative results. The Pixel Spread Model (PSM) (Li et al., 2023) uses a decoupled probabilistic model that combines the efficiency of GAN optimization with the prediction traceability of diffusion models. Latent Diffusion Models (LDM) (Rombach et al., 2022) decouple two tasks: image processing and compression, and the denoising process is learned in latent space instead of pixel space. Inpainting

is performed by encoding the masked input image, downsampling the inpainting mask, and concatenating them as additional conditions to the denoising model. Designing an image and mask conditional diffusion model requires a special architecture to accept additional input for inpainting, as done in (Rombach et al., 2022; Li et al., 2023), and often treats inpainting as a domain translation task. In contrast, TD-Paint does not require any architecture modification by directly combining the masked input and current generation with different noise levels. Differential Diffusion (Levin & Fried, 2023) introduces a new approach to soft-inpainting, where both the generated region and the conditional input are modified to ensure coherence across the entire image. This method uses LDM along with a strength map to focus on different image regions during each diffusion step. However, Differential Diffusion operates on noisy conditional inputs. The Manifold Constrained Gradient (MCG) (Chung et al., 2022) adds a correction term to ensure each sample step remains close to the data manifold, allowing for more stable inpainting. RePaint (Lugmayr et al., 2022) which aligns with our approach, combining the noisy conditional region with the current generation, where the diffusion model iteratively updates missing pixels using the surrounding context. We observe that during the early inpainting step of RePaint, the condition is dominated by noise and does not contain any semantic information. This causes the model prediction to deviate from the target manifold (see Figure 1(left)). A resampling mechanism is needed to synchronize the condition and generation regions, allowing semantically corrected images to be produced at the cost of significantly increasing computation time.

Instead of degrading the condition to the same level as the generation, we propose keeping it clean and conditioning the missing part on the known pixels from the beginning of the generation process (see Figure 1(right)). Although it requires fine-tuning the model, our approach provides much faster sampling by avoiding resampling (Lugmayr et al., 2022) or additional constraints(Chung et al., 2022). We use temporal information to model the detail in the image. The condition, containing clean data, is assigned a low noise level while the generation region starts with maximum noise that gradually decreases during the process. This approach allows for a direct combination of the clean and generated regions in the same space without changing the model architecture.

## 3 BACKGROUND ON INPAINTING DIFFUSION MODELS

**Diffusion Models:** Diffusion models learn a data distribution from a training dataset by inverting a noise process. During training, the forward diffusion process transforms a data point $x_0$ into Gaussian noise $x_T \sim \mathcal{N}(0, \mathbf{I})$ in $T$ steps by creating a series of latent variables $x_1, ..., x_T$ using:

$$q(x_t|x_{t-1}) = \mathcal{N}(x_t; \sqrt{1 - \beta_t}x_{t-1}, \beta_t\mathbf{I}), \tag{1}$$

where $\beta_t$ represents the predefined variance schedule. Given $\alpha_t = 1 - \beta_t$, $\bar{\alpha}_t = \prod_{i=1}^{t} \alpha_i$, and $\epsilon \sim \mathcal{N}(0, \mathbf{I})$, $x_t$ at step $t$ can be marginalized from $x_0$ using the reparametrization trick as follows:

$$x_t = \sqrt{\bar{\alpha}_t}x_0 + \sqrt{1 - \bar{\alpha}_t}\epsilon. \tag{2}$$

The reverse denoising process $p_\theta(x_{t-1}|x_t, t)$ allows generating from the data distribution by first sampling from $x_T \sim \mathcal{N}(0, \mathbf{I})$ and iteratively reducing the noise in the sequence $x_T, ..., x_0$. The model $\epsilon_\theta(x_t, t)$ is trained to predict the added noise $\epsilon$ to produce the sample $x_t$ at time step $t$. The model is trained using mean square error (MSE):

$$\mathcal{L} = \mathbb{E}_{\epsilon \sim \mathcal{N}(0,\mathbf{I}),x_0,t}\|\epsilon_\theta(\sqrt{\bar{\alpha}_t}x_0 + \sqrt{1 - \bar{\alpha}_t}\epsilon, t) - \epsilon\|_2^2. \tag{3}$$

**RePaint Methodology:** Given an image $x$ and a binary mask $m$, the goal of inpainting is to generate the missing region $x^\ominus$ specified by $x \odot (1 - m)$ conditioned on the known region $x^\oplus$ specified by $x \odot m$. For this, RePaint combines a noisy version of the condition $x_0 \odot m$ with the previous output of the generation process:

$$x_{t-1}^\oplus \sim \mathcal{N}\left(\sqrt{\bar{\alpha}_t}x_0, (1 - \bar{\alpha}_t)\mathbf{I}\right) \tag{4}$$

$$x_{t-1}^\ominus \sim \mathcal{N}\left(\mu_\theta(x_t, t), \Sigma_\theta(x_t, t)\right) \tag{5}$$

$$x_{t-1} = x_{t-1}^\oplus \odot m + x_{t-1}^\ominus \odot (1 - m) \tag{6}$$

RePaint uses the known pixels as a noisy condition to guide the generation of the unknown pixels. In the first inpainting step, inputs contain a high noise level and limited information about the condition. This leads to samples that deviate from the intended semantics of the condition, often resulting in

artifacts. To address this, RePaint introduces a resampling mechanism that harmonizes the two semantics by applying forward diffusion on the output $x_{t-1}$ back to $x_{t+j}$. The denoising and re-noising process involves executing the same diffusion step multiple times during the generation phase, sacrificing computational efficiency to achieve higher image quality.

## 4   THE PROPOSED TD-PAINT METHOD

In TD-Paint, the model is conditioned on known and unknown regions using the time step $t$, already present in diffusion models. Instead of using $x_{t-1}^{\oplus}$, $x_0$ is combined directly with $x_t$ without forward diffusion. This results in a faster diffusion process without changing the model architecture.

### 4.1   NOISE MODELING FOR FAST INPAINTING

**Training Process:** The objective of TD-Paint is to enable the diffusion model to discern the information content of each input region. This allows the model to differentiate between conditioned regions and those needing to be painted. By maintaining the known regions free of additional noise, TD-Paint preserves the maximum amount of information in these areas. Using the time step $t$ allows for smooth and continuous modeling of information content. Regions from the known part of the image are given a $t$ value close to zero, indicating minimal noise. On the other hand, regions that need to be inpainted start with a $t$ value close to $T$, which progressively decreases during the generation process. To do so, $t \in \{0, ..., T\}$ is transformed from a scalar to a tensor $\tau \in \{0, ..., T\}^{h \times w}$. Similarly, other variables are accommodated to perform a pixel-wise diffusion process. With one $t$ per pixel, $\alpha_t$ (resp. $\bar{\alpha}_t$, $\beta_t$) becomes $\alpha_\tau$ (resp. $\bar{\alpha}_\tau$, $\beta\tau$). This innovative modification can be integrated into most diffusion model training pipelines. Figure 2 illustrates the intermediate images used for training. During training, each input image pixel $x_{i,j}$ receives an amount of noise controlled by $\tau_{i,j}$. The forward diffusion process is then applied to $x$ on a pixel-wise basis, as illustrated in Figure 2 and can be formulated as:

$$x_\tau = \sqrt{\bar{\alpha}_\tau} x_0 + \sqrt{1 - \bar{\alpha}_\tau}\epsilon, \tag{7}$$

in which $\epsilon \sim \mathcal{N}(\mathbf{0}, \mathbf{I})$, $\tau \sim \phi_{\text{train}}$ and $\phi_{\text{train}}$ is a training strategy to sample different noise per pixel. The diffusion network then predicts $\epsilon$, using less noisy regions to reconstruct more noisy regions by optimizing the loss:

$$\mathcal{L} = \|\epsilon - \epsilon_\theta(x_\tau, \tau)\|_2^2. \tag{8}$$

The final component of TD-Paint training is the strategy used for $\phi_{\text{train}}$. A random patch size and a proportion of patches are sampled to define a condition. Based on this sampled proportion, the input image is then divided into known regions $x^{\oplus}$ and unknown regions $x^{\ominus}$ (with corresponding $t^{\oplus} = 0$ and $t^{\ominus} = t$). The possible patch sizes are defined as powers of two, i.e., $2^i | i \in \mathbb{N}, 2^i \leq w$, up to the maximum size of the image. The fraction of pixels designated as the known region is represented by a real value within the interval $[0, 1]$, ensuring that at least one patch remains in the unknown region. For example, as illustrated in Figure 2 for a patch size of 128, 25% of the pixels are assigned to the known region.

**Generation Process:** Inpainting an image with TD-Paint involves sampling a time $t$ for the conditioning region $x^{\oplus}$ and a time $t$ for the inpainted region $x^{\ominus}$ using $\phi$. Unless otherwise specified, we set $\phi_t^{\oplus} = 0$ for the condition and $\phi_t^{\ominus} = t$ for the region to inpaint for training and generation. For generation, the known region $x^{\oplus}$ is merged with the current unknown region $x_t^{\ominus}$, and one reverse step can be expressed as:

$$x_\tau = x_t^{\ominus} \odot (1 - m) + x_0^{\oplus} \odot m \tag{9}$$

$$x_{t-1}^{\ominus} \sim \mathcal{N}\left(\mu_\theta(x_\tau, \tau), \Sigma_\theta(x_\tau, \tau)\right). \tag{10}$$

This approach allows the use of input-known regions from the beginning of the diffusion process. Consequently, we can eliminate RePaint's resampling mechanism, resulting in faster inpainting. The general algorithm for inpainting an image with an arbitrary $\phi$ function is given by Algorithm 1.

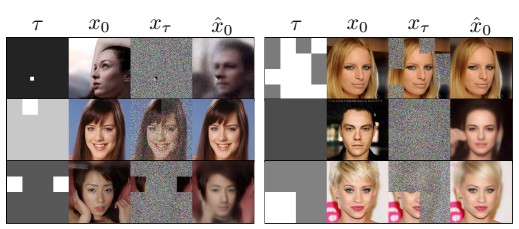

Figure 2: **Illustration of the TD-Paint patch-wise training procedure.** Each region of the input $x_0$ receives a different level of noise controlled by $\tau$. The network uses less noisy regions to reconstruct more noisy regions.

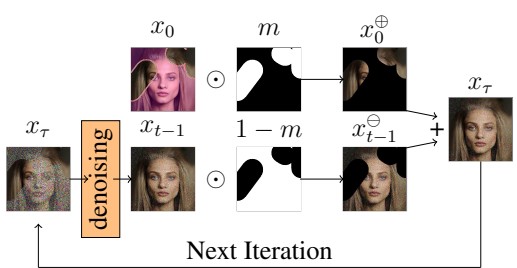

Figure 3: **The conditional generation procedure**. TD-Paint modifies the standard denoising process to condition the diffusion model on the known region without noise while gradually denoising the generated region.

## 4.2 TIME-AWARE DIFFUSION ARCHITECTURE

---

**Algorithm 1** TD-Paint Generation Process.

---

**Require:** $x^\oplus \sim q(x_0)$ a condition
**Require:** $m$ a condition mask
**Require:** $\phi_t$ giving the condition noise level for the known and unknown regions
1: $x_T \sim \mathcal{N}(\mathbf{0}, \mathbf{I})$
2: **for** $t = T, ..., 1$ **do**
3:     $\epsilon \sim \mathcal{N}(0, \mathbf{I})$
4:     $z \sim \mathcal{N}(0, \mathbf{I})$ if $t > 1$, else $z = 0$
5:     $x^\oplus_{\phi_t^\oplus} = \sqrt{\bar\alpha_{\phi_t^\oplus}} x^\oplus + \sqrt{1 - \bar\alpha_{\phi_t^\oplus}} \epsilon$
6:     $\tau = \phi_t^\ominus \odot (1 - m) + \phi_t^\oplus \odot m$
7:     $x_\tau = x^\ominus_{\phi_t^\ominus} \odot (1 - m) + x^\oplus_{\phi_t^\oplus} \odot m$
8:     $x_{t-1} = \frac{1}{\sqrt{\bar\alpha_\tau}} \left( x_\tau - \frac{\beta_\tau}{\sqrt{1 - \bar\alpha_\tau}} \epsilon_\theta(x_\tau, \tau) \right) + \sigma_\tau z$
9: **end for**
10: **return** $x_0$

---

Diffusion models using U-Net architectures (Ronneberger et al., 2015) can incorporate a time map $\tau$ without requiring architectural modifications. This adaptation is achieved by applying pixel-wise time conditioning to each pixel in the feature maps.

Classical time conditioning first uses a position embedding layer to obtain a time embedding $\gamma \in \mathbb{R}^d$ from the time step $t$:

$$\gamma = L\left[E(t) \times \sigma(L(E(t)))\right], \qquad (11)$$

where $L$ denotes linear layers, $\sigma$ is the sigmoid function, and $E$ is a sinusoidal timestep embedding. Additionally, at each layer of the U-Net, the vector $\gamma$ is used to perform time conditioning with scale-shift normalization and can be written as:

$$h^{l+1}_{i,j} = GN(h^l)_{i,j} \times (1 + L_{\text{scale}}(\gamma)) + L_{\text{shift}}(\gamma), \qquad (12)$$

where $h^l \in \mathbb{R}^{c_l \times h_l \times w_l}$ are the current features for layer $l$, $GN$ is a group normalization layer, while $L_{\text{scale}}$ and $L_{\text{shift}}$ are linear layers that change the dimension of $\gamma$ from $d$ to $c_l$. We apply the pixel-wise time conditioning across each spatial dimension $h^l$ by using $\tau_{i,j}$ instead of $t$ as follows:

$$\Gamma_{i,j} = L(E(D(\tau)_{i,j}) \times \sigma(L(E(D(\tau)_{i,j})))) \qquad (13)$$

$$h^{l+1}_{i,j} = GN(h^l)_{i,j} \times (1 + L_{\text{scale}}(\Gamma_{i,j})) + L_{\text{shift}}(\Gamma_{i,j}), \qquad (14)$$

where $\Gamma \in \mathbb{R}^{d \times h \times w}$ is the embedding of time $\tau$, and $D$ is a downscaling operation function that rescales $\tau$ to $h_l \times w_l$[1].

## 5 RESULTS AND DISCUSSION

This section empirically shows that TD-Paint: (a) produces high-quality inpainting with a clean condition on various mask sizes and shapes, on par or better than other inpainting models; (b) provides more efficient sampling, making it faster than other diffusion-based models without requiring a dedicated architecture; (c) generates diverse, high-quality images. Details on the training masks and their corresponding ablation study are presented in Appendix A, and additional qualitative results are provided in Appendix D. In Appendix C, TD-Paint is compared with state-of-the-art mask and image-conditioned inpainting models: the CNN-based LaMa (Suvorov et al., 2022) and transformer-based MAT (Li et al., 2022).

---

[1]We use bilinear interpolation, but min-pool or other techniques could be used with a similar effect.

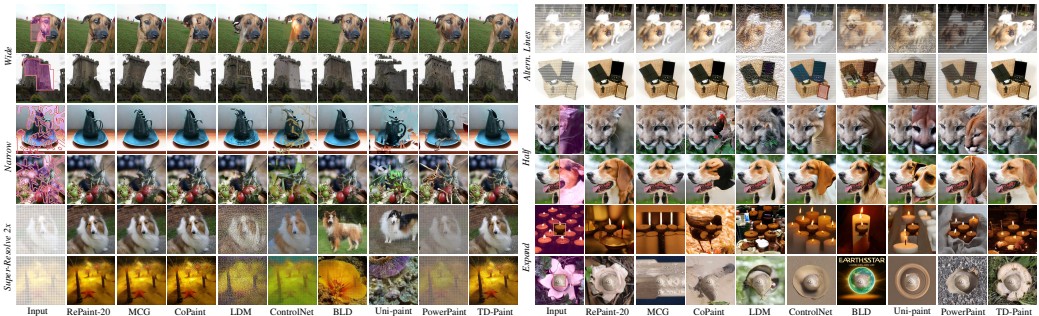

Figure 4: **Qualitative results:** TD-Paint against state-of-the-art inpainting methods on CelebA-HQ. Zoom in for better details. Additional examples can be found in Appendix D.

Figure 5: **Qualitative results:** TD-Paint against state-of-the-art inpainting methods on ImageNet1K. Additional examples can be found in Appendix D.

## 5.1 EXPERIMENTAL METHODOLOGY

**Baselines:** We evaluate TD-Paint against state-of-the-art diffusion-based inpainting methods in the pixel space: RePaint (Lugmayr et al., 2022), which conditions the generative process using noisy inputs and synchronizes them with the output through resampling. MCG (Chung et al., 2022), which adds a correction term to keep the generation closer to the data manifold, and CoPaint (Zhang et al., 2023a) that utilizes Tweedie's formula for better generation. Recent studies (Chung et al., 2022; Zhang et al., 2023a) have shown that CoPaint and MCG are among the best-performing inpainting methods. We denote the RePaint-20 model as using 20 resampling steps, while RePaint-1 refers to the model with 1 resampling step matching the number of steps used by TD-Paint. Furthermore, we conduct comparisons with latent diffusion models that have access to the complete context: LDM (Rombach et al., 2022) and ControlNet (Zhang et al., 2023b). Additionally, we compare against foundation model-based latent diffusion approaches, including Blended Latent Diffusion (BLD) (Avrahami et al., 2023), Uni-paint (Yang et al., 2023), and PowerPaint (Zhuang et al., 2025).

**Implementation Details:** Our approach is validated using the CelebA-HQ (Karras et al., 2018) dataset, the ImageNet1K (Russakovsky et al., 2015) dataset, and the Places2 dataset (Zhou et al., 2018) at 256x256 resolution. We modify the implementation of (Dhariwal & Nichol, 2021), maintaining all their hyperparameters. Training on CelebA-HQ is conducted for approximately 150K steps with batch size 64 on 4 A100, for ImageNet1K and Places2 for about 200K steps with batch size 128 on 8 A100. For baselines, we utilize existing code and pre-trained models when available. For ImageNet1K, we train LaMa for 1M steps on batch size 5 using their implementation, and MAT for 300K steps on batch size 32 using their implementation. Both LDM and ControlNet are trained using computational resources equivalent to TD-Paint. For LDM, the encoded masked image and the downsampled mask provide additional context during the sampling process. For ControlNet, the encoded masked image alone provides additional context.

**Evaluation Metrics:** Image quality is evaluated using established metrics from the inpainting literature: the Learned Perceptual Image Patch Similarity (Zhang et al., 2018) (LPIPS), the Structural Similarity Index Measure (Wang et al., 2004) (SSIM), and the Kernel Inception Distance (Bińkowski et al., 2018) (KID) (using the TorchMetrics (Nicki Skafte Detlefsen et al., 2022) implementation). The number of diffusion steps (NBS) and the mean time to inpaint an image (Runtime) are used to evaluate the computational efficiency of TD-Paint. For evaluation, we use 2,824 images from the CelebA-HQ test set, 5,000 images from ImageNet1K, and 2,000 images from Places2.

Figure 6: **Qualitative results:** TD-Paint against state-of-the-art inpainting methods on Places2. Additional examples can be found in Appendix D.

## 5.2 COMPARISON WITH DIFFUSION-BASED MODELS

***Wide* and *Narrow* masks** In the standard image inpainting scenario, TD-Paint is compared using *Wide* and *Narrow* masks following LaMa (Suvorov et al., 2022) protocol. Tables 1 and 2 shows that TD-Paint consistently outperforms other diffusion-based models, improving by 20% RePaint-20 LPIPS's on the *Wide* mask on CelebA-HQ, ImageNet1K and Places2, and by 30% on the *Narrow* masks. MCG lacks global consistency, resulting in significant artifacts on *Wide* masks, as seen in Figures 4 and 5 where it produces eyes of different colors, strange textures, and inpainting artifacts. RePaint produces high-quality images at the cost of significant inference time. Our approach can produce high-quality images while requiring much less processing time. When processing small masks, LDM occasionally produces minor artifacts, as evident in the top rows of Figures 4 and 5. These artifacts are more pronounced in ControlNet, which lacks mask information, resulting in inconsistencies between known and unknown regions. Among foundation-based models, BLD demonstrates superior performance across all datasets, while Uni-paint and PowerPaint show inferior results on CelebA-HQ but achieving better performance on ImageNet1K and Places2. This performance variation may be attributed to BLD's use of the provided LDM baselines (our early experiments with Stable Diffusion yielded less favorable results), whereas both Uni-paint and PowerPaint utilize Stable Diffusion as their foundation.

***Super-Resolve 2x* and *Altern. Lines* masks** In the *Super-Resolve 2x* setting, every other pixel is removed from the image, while the *Altern. Lines* setting removes every other line. All pixel-based baselines achieve low LPIPS scores and produce high-quality images for both types of masks (see Figures 4 to 6). In contrast, latent-based models prove inadequate for this task due to their downsampling of inpainting masks, which results in critical information loss. TD-Paint outperforms all considered baselines on CelebA-HQ and comes a close second to CoPaint on ImageNet1K and Places2, as shown in Tables 1 and 2. The third-best performing model is RePaint. Compared to RePaint, TD-Paint improves LPIPS by 115% in the *Super-Resolve 2x* setting and by 69% in the *Altern. Lines* setting.

***Half* and *Expand* masks** The *Half* setting removes the right part of the images, and the *Expand* setting keeps only the central 64×64 part of a 256×256 pixel image. Both LPIPS and SSIM metrics are less suitable when a significant portion of the image is missing, as they rely on a single ground truth. This penalizes methods that generate realistic images with semantics different from the ground truth (Lugmayr et al., 2022). In this case, the KID, which measures the distance between distributions, is more reliable for assessing image quality. Applying *Half* and *Expand* masks is particularly challenging, as they remove substantial portions of the images. task where an important part of the images is removed. This complexity is shown both visually in Figures 4 to 6 and by the quantitative results in Tables 1 and 2. Our model performs best on both the *Half* and *Expand* masks across all datasets (except for CelebA-HQ where it comes close to LDM), as measured by the KID, while significantly reducing the inference time. Figure 4 shows that TD-Paint can produce high-quality images in this challenging setting where RePaint lacks global consistency. On ImageNet1K and Places2 (see Figures 5 and 6), we observe that MCG and CoPaint often mirror the images from the *Half* mask, resulting in symmetrical outputs. Similar behavior is observed on *Expand* masks, with significant texture blending. ControlNet exhibits limited generalization capability when handling very large masks, likely due to the absence of mask information, whereas LDM demonstrates robust performance in these scenarios. Our analysis reveals that foundation-based models yield inferior qualitative results on CelebA-HQ and Places2 datasets. However, they show improved performance on ImageNet1K, where class names provide supplementary textual information, which is particularly beneficial in scenarios with limited contextual information.

Table 1: **Quantitative results:** LPIPS and SSIM evaluation of diffusion models for inpainting on the CelebA-HQ, ImageNet1K and Places2 datasets.

| CelebA-HQ | Wide | | Narrow | | Super-Resolve 2x | | Altern. Lines | | Half | | Expand | |
|---|---|---|---|---|---|---|---|---|---|---|---|---|
| | LPIPS↓ | SSIM↑ | LPIPS↓ | SSIM↑ | LPIPS↓ | SSIM↑ | LPIPS↓ | SSIM↑ | LPIPS↓ | SSIM↑ | LPIPS↓ | SSIM↑ |
| RePaint-1 | 0.098 | 0.823 | 0.076 | 0.857 | 0.273 | 0.680 | 0.046 | 0.925 | 0.230 | 0.581 | 0.568 | 0.133 |
| RePaint-20 | 0.067 | 0.864 | 0.036 | 0.906 | 0.037 | 0.904 | 0.016 | 0.951 | 0.189 | 0.645 | 0.489 | 0.191 |
| MCG | 0.070 | 0.823 | 0.045 | 0.856 | 0.081 | 0.829 | 0.030 | 0.903 | 0.173 | 0.639 | 0.437 | 0.250 |
| CoPaint | 0.073 | 0.835 | 0.044 | 0.877 | 0.033 | 0.899 | 0.020 | 0.929 | 0.185 | 0.638 | 0.454 | 0.210 |
| LDM | 0.060 | 0.863 | 0.049 | 0.879 | 1.233 | 0.040 | 0.740 | 0.126 | **0.168** | 0.662 | **0.432** | 0.228 |
| ControlNet | 0.091 | 0.834 | 0.199 | 0.763 | 0.593 | 0.187 | 0.318 | 0.362 | 0.205 | 0.620 | 0.552 | 0.173 |
| BLD | **0.014** | **0.942** | **0.017** | **0.933** | 0.639 | 0.206 | 0.558 | 0.254 | 0.232 | 0.621 | 0.490 | **0.263** |
| Uni-paint | 0.119 | 0.805 | 0.169 | 0.776 | 0.825 | 0.072 | 0.711 | 0.127 | 0.282 | 0.555 | 0.687 | 0.102 |
| PowerPaint | 0.111 | 0.821 | 0.110 | 0.837 | 0.788 | 0.111 | 0.567 | 0.207 | 0.300 | 0.548 | 0.604 | 0.155 |
| TD-Paint | 0.055 | 0.873 | 0.028 | 0.918 | **0.017** | **0.971** | **0.010** | **0.971** | 0.170 | **0.667** | 0.457 | 0.212 |
| **ImageNet1K** | Wide | | Narrow | | Super-Resolve 2x | | Altern. Lines | | Half | | Expand | |
| RePaint-1 | 0.169 | 0.781 | 0.137 | 0.794 | 0.622 | 0.268 | 0.245 | 0.619 | 0.336 | 0.542 | 0.680 | 0.097 |
| RePaint-20 | 0.124 | 0.820 | 0.067 | 0.854 | 0.169 | 0.693 | 0.089 | 0.816 | 0.301 | 0.590 | 0.676 | 0.134 |
| MCG | 0.120 | 0.780 | 0.075 | 0.806 | 0.182 | 0.649 | 0.104 | 0.768 | 0.273 | 0.561 | 0.634 | 0.152 |
| CoPaint | 0.137 | 0.798 | 0.080 | 0.835 | **0.071** | **0.818** | **0.040** | **0.884** | 0.298 | 0.578 | 0.645 | 0.145 |
| LDM | 0.138 | 0.744 | 0.117 | 0.748 | 1.101 | 0.032 | 0.687 | 0.102 | 0.292 | 0.538 | 0.620 | 0.135 |
| ControlNet | 0.171 | 0.728 | 0.234 | 0.672 | 0.605 | 0.163 | 0.340 | 0.340 | 0.326 | 0.523 | 0.686 | 0.115 |
| BLD | **0.050** | **0.831** | 0.054 | 0.819 | 0.737 | 0.134 | 0.599 | 0.190 | 0.356 | 0.521 | 0.666 | **0.172** |
| Uni-paint | 0.210 | 0.694 | 0.263 | 0.645 | 0.795 | 0.076 | 0.633 | 0.153 | 0.365 | 0.480 | 0.705 | 0.082 |
| PowerPaint | 0.196 | 0.712 | 0.182 | 0.725 | 0.797 | 0.097 | 0.520 | 0.217 | 0.356 | 0.490 | 0.657 | 0.100 |
| TD-Paint | 0.099 | 0.830 | 0.057 | **0.864** | 0.136 | 0.648 | 0.059 | 0.847 | **0.257** | **0.603** | **0.597** | 0.159 |
| **Places2** | Wide | | Narrow | | Super-Resolve 2x | | Altern. Lines | | Half | | Expand | |
| RePaint-1 | 0.179 | 0.776 | 0.152 | 0.788 | 0.544 | 0.332 | 0.232 | 0.656 | 0.347 | 0.541 | 0.696 | 0.090 |
| RePaint-20 | 0.138 | 0.816 | 0.078 | 0.853 | 0.155 | 0.729 | 0.085 | 0.841 | 0.320 | 0.584 | 0.688 | 0.121 |
| MCG | 0.131 | 0.788 | 0.092 | 0.809 | 0.250 | 0.626 | 0.115 | 0.786 | **0.269** | 0.577 | 0.618 | 0.156 |
| CoPaint | 0.133 | 0.804 | 0.082 | 0.842 | **0.071** | **0.828** | **0.037** | **0.902** | 0.282 | 0.584 | 0.630 | 0.150 |
| LDM | 0.128 | 0.807 | 0.112 | 0.803 | 1.176 | 0.026 | 0.706 | 0.091 | 0.283 | 0.582 | 0.612 | 0.142 |
| ControlNet | 0.173 | 0.781 | 0.234 | 0.730 | 0.614 | 0.170 | 0.312 | 0.392 | 0.327 | 0.548 | 0.670 | 0.102 |
| BLD | **0.035** | **0.901** | **0.039** | **0.886** | 0.744 | 0.134 | 0.600 | 0.206 | 0.335 | 0.567 | 0.663 | **0.173** |
| Uni-paint | 0.184 | 0.753 | 0.219 | 0.717 | 0.835 | 0.071 | 0.651 | 0.159 | 0.344 | 0.522 | 0.727 | 0.082 |
| PowerPaint | 0.187 | 0.774 | 0.152 | 0.793 | 0.797 | 0.097 | 0.483 | 0.258 | 0.348 | 0.536 | 0.653 | 0.101 |
| TD-Paint | 0.112 | 0.826 | 0.064 | 0.865 | 0.130 | 0.696 | 0.060 | 0.879 | 0.273 | **0.594** | **0.607** | 0.146 |

## 5.3 QUALITY VS. EFFICIENCY

We compare the time efficiency of different diffusion approaches working in the pixel space by computing the average time to sample 100 images consecutively on a single V100, and the results are reported in Table 3. State-of-the-art approaches require over $6\times$ longer to sample compared to TD-Paint. The increased time in RePaint is due to the resampling mechanism needed to synchronize condition and generation, while MCG requires an additional backward pass and more steps to optimize the image.

Table 3: Inpainting speed for different diffusion model in the pixel space.

| CelebA-HQ | LPIPS↓ | Runtime↓ | NBS↓ |
|---|---|---|---|
| RePaint-1 | 0.076 | 19.68 | 250 |
| RePaint-20 | 0.036 | 189.62 | 4750 |
| MCG | 0.045 | 184.59 | 1000 |
| CoPaint | 0.044 | 128.14 | 1000 |
| TD-Paint | 0.028 | 30.67 | 250 |

In contrast, our model reduces inference time by trading fine-tuning costs, enabling faster generation of high-quality images than other diffusion-based inpainting models.

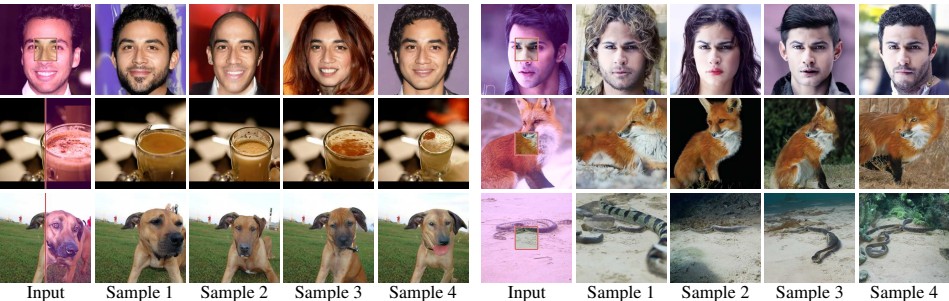

| | | | | | | | | | |
|---|---|---|---|---|---|---|---|---|---|
| Input | Sample 1 | Sample 2 | Sample 3 | Sample 4 | Input | Sample 1 | Sample 2 | Sample 3 | Sample 4 |

Figure 7: Examples of diverse generations using TD-Paint on the CelebA and ImageNet, with the same input image and different initial noise. Additional examples are available in Appendix D.

Table 2: **Quantitative results:** KID evaluation of diffusion models for inpainting on the CelebA-HQ, ImageNet1K and Places2 datasets.

| CelebA-HQ | KID↓ | | | | | |
|---|---|---|---|---|---|---|
| | *Wide* | *Narrow* | *Super-Res. 2x* | *Altern. Lines* | *Half* | *Expand* |
| RePaint-1 | 0.001 62 | 0.004 23 | 0.072 77 | 0.012 71 | 0.004 34 | 0.023 99 |
| RePaint-20 | 0.001 15 | 0.001 38 | 0.016 07 | 0.006 15 | 0.004 18 | 0.024 15 |
| MCG | 0.001 10 | 0.001 49 | 0.022 92 | 0.004 93 | 0.001 18 | 0.010 96 |
| CoPaint | 0.001 97 | 0.002 23 | 0.007 48 | 0.003 08 | 0.003 42 | 0.011 03 |
| LDM | 0.000 10 | 0.001 85 | 0.304 01 | 0.236 88 | 0.001 01 | **0.004 96** |
| ControlNet | 0.003 14 | 0.099 10 | 0.180 62 | 0.072 80 | 0.003 57 | 0.047 67 |
| BLD | −0.000 05 | 0.000 42 | 0.134 01 | 0.103 56 | 0.005 20 | 0.025 67 |
| Uni-paint | 0.007 68 | 0.054 62 | 0.450 69 | 0.252 60 | 0.010 02 | 0.144 14 |
| PowerPaint | 0.008 30 | 0.018 16 | 0.321 38 | 0.146 03 | 0.051 07 | 0.217 72 |
| TD-Paint | **−0.000 08** | **−0.000 09** | **0.000 59** | **0.000 24** | **0.000 44** | 0.007 10 |
| **ImageNet1K** | *Wide* | *Narrow* | *Super-Res. 2x* | *Altern. Lines* | *Half* | *Expand* |
| RePaint-1 | 0.001 28 | 0.001 51 | 0.111 27 | 0.010 39 | 0.002 24 | 0.004 05 |
| RePaint-20 | 0.000 16 | −0.000 07 | 0.003 64 | 0.000 68 | 0.001 06 | 0.005 94 |
| MCG | 0.000 78 | 0.000 07 | 0.004 23 | 0.001 28 | 0.007 41 | 0.053 79 |
| CoPaint | 0.004 57 | 0.000 34 | **0.000 39** | **0.000 05** | 0.015 32 | 0.085 38 |
| LDM | 0.009 31 | 0.007 89 | 0.215 63 | 0.150 21 | 0.018 03 | 0.076 00 |
| ControlNet | 0.005 34 | 0.014 42 | 0.063 52 | 0.017 14 | 0.003 97 | 0.005 42 |
| BLD | 0.005 44 | 0.005 89 | 0.043 20 | 0.064 94 | 0.010 03 | 0.013 78 |
| Uni-paint | 0.007 08 | 0.022 50 | 0.064 00 | 0.092 72 | 0.006 11 | 0.010 79 |
| PowerPaint | 0.006 54 | 0.008 57 | 0.157 53 | 0.055 63 | 0.006 33 | 0.009 80 |
| TD-Paint | **−0.000 01** | **−0.000 10** | 0.004 72 | 0.000 41 | **0.000 28** | **0.003 86** |
| **Places2** | *Wide* | *Narrow* | *Super-Res. 2x* | *Altern. Lines* | *Half* | *Expand* |
| RePaint-1 | 0.003 11 | 0.002 21 | 0.110 56 | 0.011 34 | 0.014 52 | 0.025 65 |
| RePaint-20 | 0.000 44 | −0.000 26 | 0.004 58 | 0.001 47 | 0.006 15 | 0.032 35 |
| MCG | 0.000 35 | 0.000 47 | 0.009 92 | 0.002 96 | 0.004 98 | 0.016 09 |
| CoPaint | 0.000 31 | −0.000 05 | **0.000 76** | **0.000 16** | 0.001 84 | 0.020 94 |
| LDM | 0.000 78 | 0.001 37 | 0.237 59 | 0.139 34 | 0.003 92 | 0.018 58 |
| ControlNet | 0.002 77 | 0.009 95 | 0.126 10 | 0.023 21 | 0.008 52 | 0.018 70 |
| BLD | **−0.000 25** | −0.000 09 | 0.123 99 | 0.121 02 | 0.018 62 | 0.033 62 |
| Uni-paint | 0.001 57 | 0.009 01 | 0.241 63 | 0.165 31 | 0.004 65 | 0.026 02 |
| PowerPaint | 0.018 14 | 0.005 83 | 0.217 82 | 0.057 73 | 0.074 71 | 0.251 31 |
| TD-Paint | −0.000 16 | **−0.000 30** | 0.005 25 | 0.000 27 | **0.000 64** | **0.008 36** |

## 5.4 DIVERSITY OF GENERATED IMAGES

While TD-Paint performs fast and high-quality inpainting, we must ask whether this comes at the cost of diversity. To evaluate this, we compute the Diversity Score (Lugmayr et al., 2021) by generating 10 different images for 100 inputs. The quantitative results reported in Table 4. The most diverse model is RePaint-1, which also has a high LPIPS score. In contrast, TD-Paint achieves a high Diversity Score across most masks while consistently producing high-quality images (see Figure 7) with lower LPIPS scores.

Table 4: Diversity Score on CelebA-HQ with 10 random generations across 100 images.

| CelebA-HQ | Diversity Score↑ | | | | | |
|---|---|---|---|---|---|---|
| | *Wide* | *Narrow* | *Super-Resolve 2x* | *Altern. Lines* | *Half* | *Expand* |
| RePaint-1 | 23.716 | 33.355 | 32.564 | 30.917 | 22.682 | 19.669 |
| RePaint-20 | 23.003 | 28.276 | 23.296 | 23.277 | 23.059 | 22.618 |
| MCG | 17.694 | 18.084 | 17.427 | 17.547 | 17.690 | 17.369 |
| CoPaint | 23.979 | 27.586 | 24.840 | 24.279 | 23.802 | 23.211 |
| TD-Paint | 22.629 | 29.402 | 27.358 | 29.331 | 23.459 | 21.753 |

## 6 CONCLUSIONS

In this paper, we introduce Time-aware Diffusion Paint (TD-Paint), a method that accelerates inpainting by modeling multiple noise levels through time conditioning in the diffusion process. Unlike other diffusion-based models, TD-Paint does not require any special architecture for inpainting, and generates high-quality, diverse images more quickly. This efficiency makes TD-Paint highly practical for real-world applications and usable for resource-constrained devices.

ACKNOWLEDGMENTS

This work was financially supported by the ANR Labcom LLisa ANR-20-LCV1-0009. We thank CRIANN, who provided us with the computation resources necessary for our experiments. This work was performed using HPC resources from GENCI–IDRIS (Grant 2024-AD011013862R1). We thank the Natural Sciences and Engineering Research Council of Canada (NSERC) and the Digital Research Alliance of Canada.

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

## A  ABLATIONS: TRAINING MASK

We examine the contribution of masks during training on the CelebA-HQ dataset using our mask strategy (Ours) described in Section 4.1, LaMa masks (LAMA), and a random mix of the two (Ours+LAMA). Unless stated otherwise, all results are reported with Ours+LAMA. Table 5 shows that using LAMA over Ours reduces the error for every test mask except *Super-Resolve 2x* and *Altern. Lines*, which is explained by the more complicated design of LAMA, which produces train masks closer to the *Wide* and *Narrow* test masks and of more diverse shapes. Notably, using Ours+LAMA masks still allows for very low *Super-Resolve 2x* and *Altern. Lines* errors compared to LaMa errors on the same masks in Table 6. Using Ours+LAMA allows the benefits of Ours masks to be retrained while having low *Super-Resolve 2x* and *Altern. Lines* errors.

Table 5: **Ablation study for the types of training time map.** The metrics show how the use of Ours focuses more on very fine inpainting masks. The distribution of LAMA masks is closer to larger inpainting masks, such as *Wide*. Combining the two allows for strong performance across the range of test masks considered.

| CelebA-HQ | Wide | | Narrow | | Super-Resolve 2x | | Altern. Lines | | Half | | Expand | |
| TD-Paint | LPIPS↓ | SSIM↑ | LPIPS↓ | SSIM↑ | LPIPS↓ | SSIM↑ | LPIPS↓ | SSIM↑ | LPIPS↓ | SSIM↑ | LPIPS↓ | SSIM↑ |
|---|---|---|---|---|---|---|---|---|---|---|---|---|
| Ours | 0.067 | 0.862 | 0.032 | 0.913 | 0.018 | 0.942 | 0.009 | 0.972 | 0.174 | 0.666 | 0.463 | 0.244 |
| LAMA | 0.053 | 0.876 | 0.027 | 0.920 | 0.055 | 0.870 | 0.044 | 0.920 | 0.168 | 0.665 | 0.445 | 0.229 |
| Ours+LAMA | 0.055 | 0.873 | 0.028 | 0.918 | 0.017 | 0.939 | 0.010 | 0.971 | 0.170 | 0.667 | 0.457 | 0.212 |

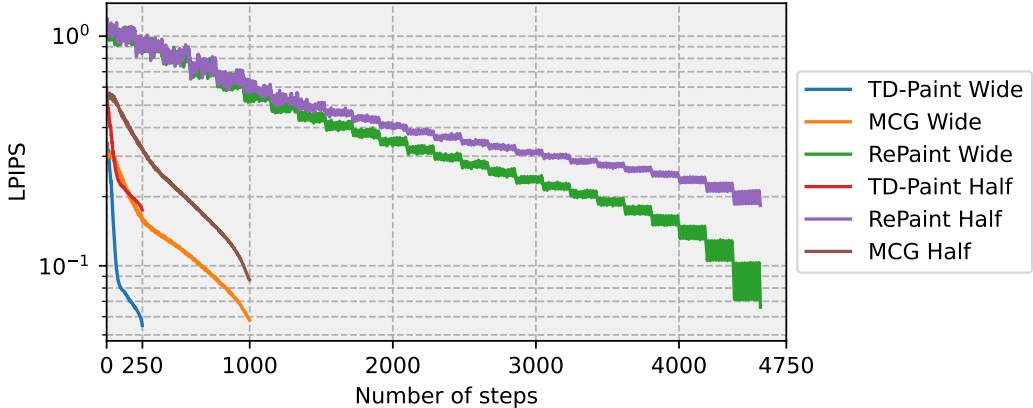

Figure 8: Image quality at different time step for 100 images on CelebA-HQ dataset for *Wide* and *Half* masks.

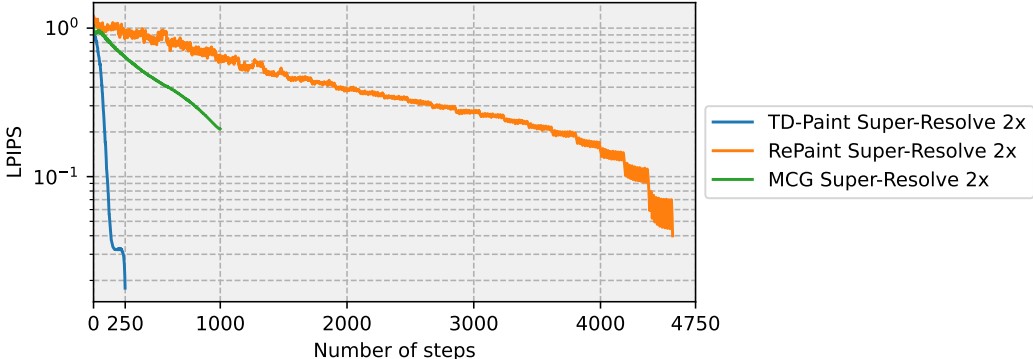

Figure 9: Image quality at different time step for 100 images on CelebA-HQ dataset for *Super-Resolve 2x* mask.

## B    IMAGE QUALITY AND DIFFUSION STEPS

We compare the LPIPS metrics over the diffusion step for TD-Paint, RePaint and MCG in Figures 8 and 9 averaged over 100 images from the CelebA-HQ dataset. TD-Paint can produce high-quality images in a fraction of the steps required by RePaint because it takes advantage of the state since the early step of the diffusion process.

## C    COMPARISON WITH CNN- AND TRANSFORMER-BASED MODELS

We compare TD-Paint with LaMa (CNN-based) and MAT (transformer-based) models in Tables 6 and 7.

***Wide* and *Narrow* masks** Our approach closely matches the performance of LaMa and MAT in the *Wide* setting on CelebA-HQ and even surpasses them in the *Narrow* setting on CelebA-HQ, as well as in both the *Wide* and *Narrow* settings on ImageNet1K. As shown in Figure 10, LaMa tends to generate pupils of different sizes when one eye is hidden in the *Wide* and different eye colors in the *Narrow* settings.

***Super-Resolve 2x* and *Altern. Lines* masks** Table 6 shows that TD-Paint outperforms the baselines by a wide margin. Particularly MAT struggles with this task and often produces images with significant artifacts and blurring (see Figures 10 to 12). In contrast, TD-Paint consistently achieves lower LPIPS scores and higher SSIM values, reflecting superior image quality and performance.

*Half* and *Expand* **masks** On CelebA-HQ, TD-Paint achieves the best results across all datasets, as indicated by the KID metrics (see Table 7). As shown in Figures 10 to 12, LaMa generates blurry artifacts in both the *Half* and *Expand* settings, whereas our proposed model consistently produces high-quality images. This behavior of LaMa may be due to overfitting to the training mask distribution, as suggested by previous studies (Lugmayr et al., 2022).

# D  ADDITIONAL QUALITATIVE RESULTS

We provide additional qualitative inpainting results compared to the state-of-the-art models described in Section 5 and Appendix C.

For CelebA-HQ on *Wide* and *Narrow* masks in Figure 13, *Super-Resolve 2x* and *Altern. Lines* masks in Figure 14, *Half* and *Expand* in Figure 15.

For ImageNet1K on *Wide* and *Narrow* masks in Figure 16, *Super-Resolve 2x* and *Altern. Lines* masks in Figure 17, *Half* and *Expand* in Figure 18.

For Places2 on *Wide* and *Narrow* masks in Figure 19, *Super-Resolve 2x* and *Altern. Lines* masks in Figure 20, *Half* and *Expand* in Figure 21.

Additional diversity results in CelebA-HQ and ImageNet1K can be found in Figure 22, and for ImageNet1K with different conditional classes in Figure 23. Through class conditioning, TD-Paint

Table 6: **Quantitative results:** evaluation of CNN- and transformer-based models for inpainting on the CelebA-HQ, ImageNet1K and Places2 datasets.

| CelebA-HQ | Wide | | Narrow | | Super-Resolve 2x | | Altern. Lines | | Half | | Expand | |
|---|---|---|---|---|---|---|---|---|---|---|---|---|
| | LPIPS↓ | SSIM↑ | LPIPS↓ | SSIM↑ | LPIPS↓ | SSIM↑ | LPIPS↓ | SSIM↑ | LPIPS↓ | SSIM↑ | LPIPS↓ | SSIM↑ |
| LaMa | **0.052** | **0.882** | 0.033 | 0.911 | 0.219 | 0.662 | 0.110 | 0.728 | **0.161** | **0.693** | **0.410** | **0.257** |
| MAT | 0.055 | 0.872 | 0.030 | 0.911 | 0.509 | 0.201 | 0.251 | 0.668 | 0.171 | 0.668 | 0.475 | 0.192 |
| TD-Paint | 0.055 | 0.873 | **0.028** | **0.918** | **0.017** | **0.939** | **0.010** | **0.971** | 0.170 | 0.667 | 0.457 | 0.212 |
| **ImageNet1K** | Wide | | Narrow | | Super-Resolve 2x | | Altern. Lines | | Half | | Expand | |
| | LPIPS↓ | SSIM↑ | LPIPS↓ | SSIM↑ | LPIPS↓ | SSIM↑ | LPIPS↓ | SSIM↑ | LPIPS↓ | SSIM↑ | LPIPS↓ | SSIM↑ |
| LaMa | 0.107 | **0.832** | 0.068 | 0.853 | 0.375 | 0.422 | 0.271 | 0.483 | 0.281 | **0.618** | 0.626 | **0.196** |
| MAT | 0.143 | 0.751 | 0.095 | 0.781 | 0.512 | 0.257 | 0.410 | 0.422 | 0.308 | 0.542 | 0.633 | 0.144 |
| TD-Paint | **0.099** | 0.830 | **0.057** | **0.864** | **0.136** | **0.648** | **0.059** | **0.847** | **0.257** | 0.603 | **0.597** | 0.159 |
| **Places2** | Wide | | Narrow | | Super-Resolve 2x | | Altern. Lines | | Half | | Expand | |
| | LPIPS↓ | SSIM↑ | LPIPS↓ | SSIM↑ | LPIPS↓ | SSIM↑ | LPIPS↓ | SSIM↑ | LPIPS↓ | SSIM↑ | LPIPS↓ | SSIM↑ |
| LaMa | **0.106** | **0.836** | 0.064 | 0.864 | 0.477 | 0.348 | 0.187 | 0.605 | 0.281 | **0.625** | 0.611 | **0.212** |
| MAT | 0.131 | 0.792 | 0.079 | 0.832 | 0.186 | 0.658 | 0.087 | 0.795 | 0.284 | 0.580 | 0.651 | 0.144 |
| TD-Paint | 0.112 | 0.826 | **0.064** | **0.865** | **0.130** | **0.696** | **0.060** | **0.879** | **0.273** | 0.594 | **0.607** | 0.146 |

Table 7: **Quantitative results:** KID evaluation of CNN- and transformer-based models for inpainting on the CelebA-HQ, ImageNet1K and Places2 datasets.

| CelebA-HQ | | | KID↓ | | | |
|---|---|---|---|---|---|---|
| | Wide | Narrow | Super-Res. 2x | Altern. Lines | Half | Expand |
| LaMa | 0.000 96 | 0.001 64 | 0.059 09 | 0.037 79 | 0.006 27 | 0.090 38 |
| MAT | 0.000 07 | 0.000 16 | 0.149 92 | 0.054 72 | 0.000 80 | 0.050 62 |
| TD-Paint | **−0.000 08** | **−0.000 09** | **0.000 59** | **0.000 24** | **0.000 44** | **0.007 10** |
| **ImageNet1K** | | | KID↓ | | | |
| | Wide | Narrow | Super-Res. 2x | Altern. Lines | Half | Expand |
| LaMa | 0.002 53 | 0.000 84 | 0.066 59 | 0.023 76 | 0.006 45 | 0.072 49 |
| MAT | 0.010 75 | 0.007 04 | 0.086 54 | 0.045 74 | 0.022 62 | 0.095 40 |
| TD-Paint | **−0.000 01** | **−0.000 10** | **0.004 72** | **0.000 41** | **0.000 28** | **0.003 86** |
| **Places2** | | | KID↓ | | | |
| | Wide | Narrow | Super-Res. 2x | Altern. Lines | Half | Expand |
| LaMa | 0.000 38 | 0.000 21 | 0.096 99 | 0.013 40 | 0.003 78 | 0.029 88 |
| MAT | 0.000 89 | 0.000 15 | 0.015 16 | 0.002 95 | 0.003 20 | 0.136 94 |
| TD-Paint | **−0.000 16** | **−0.000 30** | **0.005 25** | **0.000 27** | **0.000 64** | **0.008 36** |

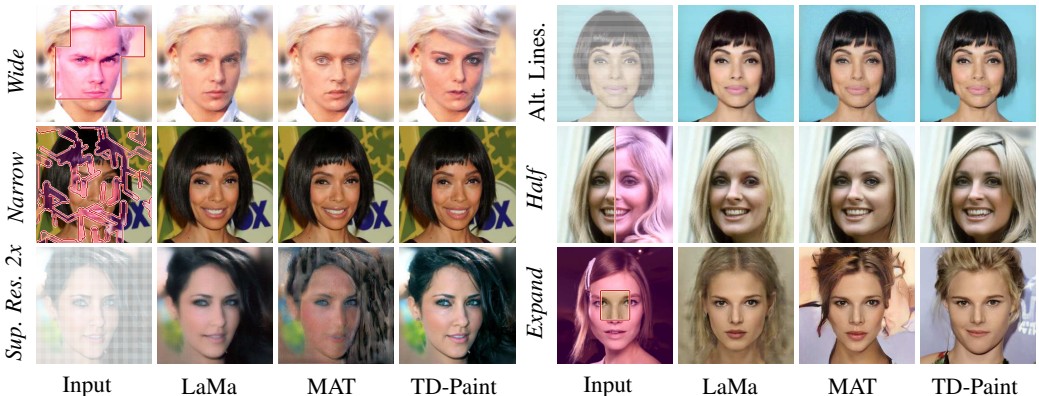

Figure 10: **Qualitative results:** TD-Paint against state-of-the-art inpainting CNN- and transformer-based models on CelebA-HQ. Zoom in for better details. Additional examples can be found in Appendix D.

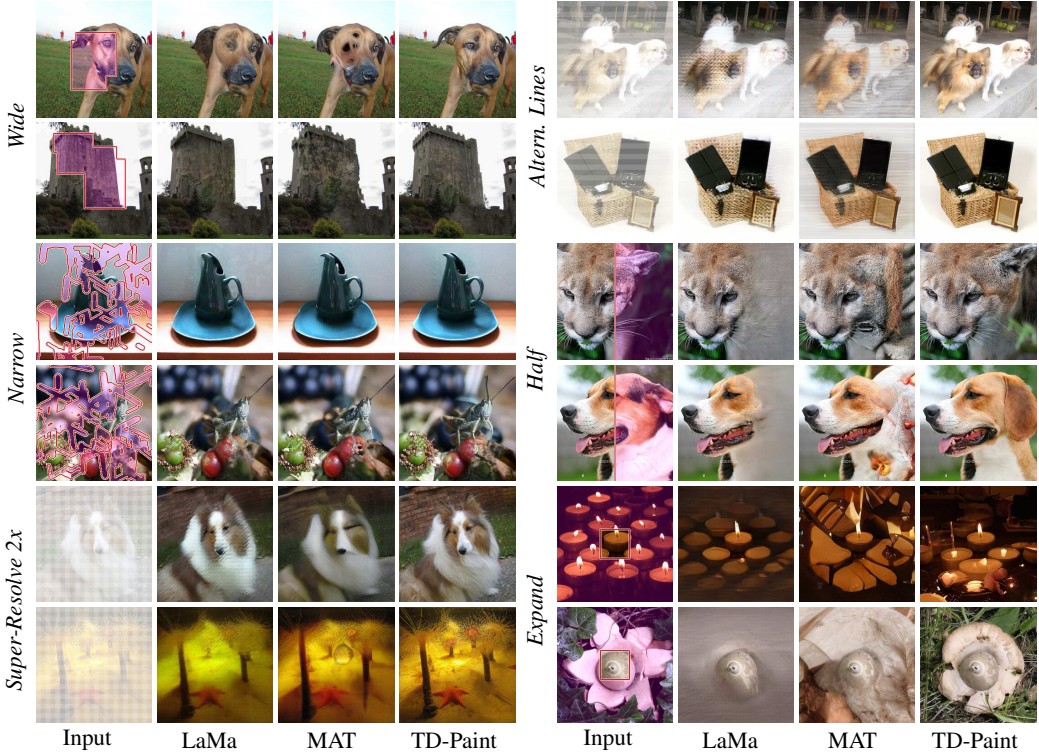

Figure 11: **Qualitative results:** TD-Paint against state-of-the-art inpainting CNN- and transformer-based models on ImageNet1K. Additional examples can be found in Appendix D.

can generate diverse images based on different target classes. This mechanism allows TD-Paint guiding the inpainting process toward specific semantic categories, resulting in more controlled and contextually relevant image completions. This feature is especially useful when the inpainted region must match a particular class, increasing TD-Paint's flexibility and effectiveness across a range of inpainting tasks.

Additional examples of inpainting using object-focused and region-specific masks are presented in Figures 24 and 25. These examples feature user-drawn masks that naturally follow object boundaries and regional structures, demonstrating TD-Paint's effectiveness in practical image manipulation scenarios. Such realistic mask shapes better reflect how users interact with inpainting tools

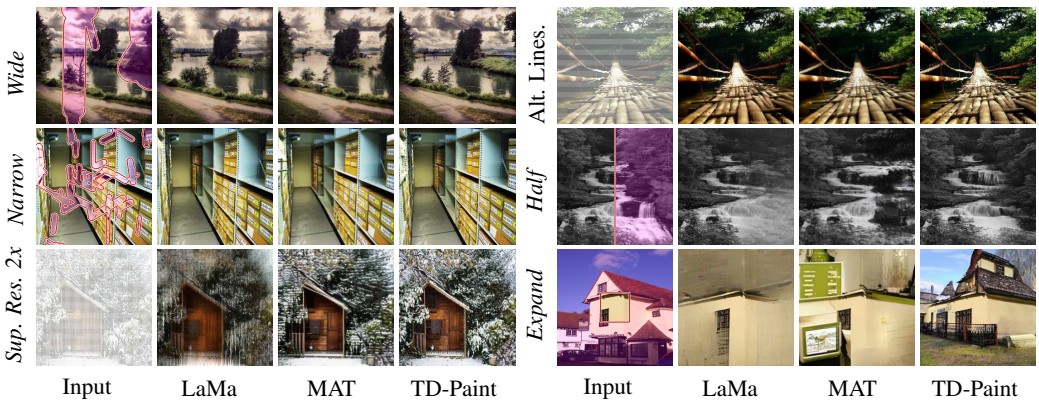

Figure 12: **Qualitative results:** TD-Paint against state-of-the-art inpainting CNN- and transformer-based models on Places2. Additional examples can be found in Appendix D.

in real-world applications, where selections typically correspond to meaningful objects or regions rather than arbitrary geometric patterns.

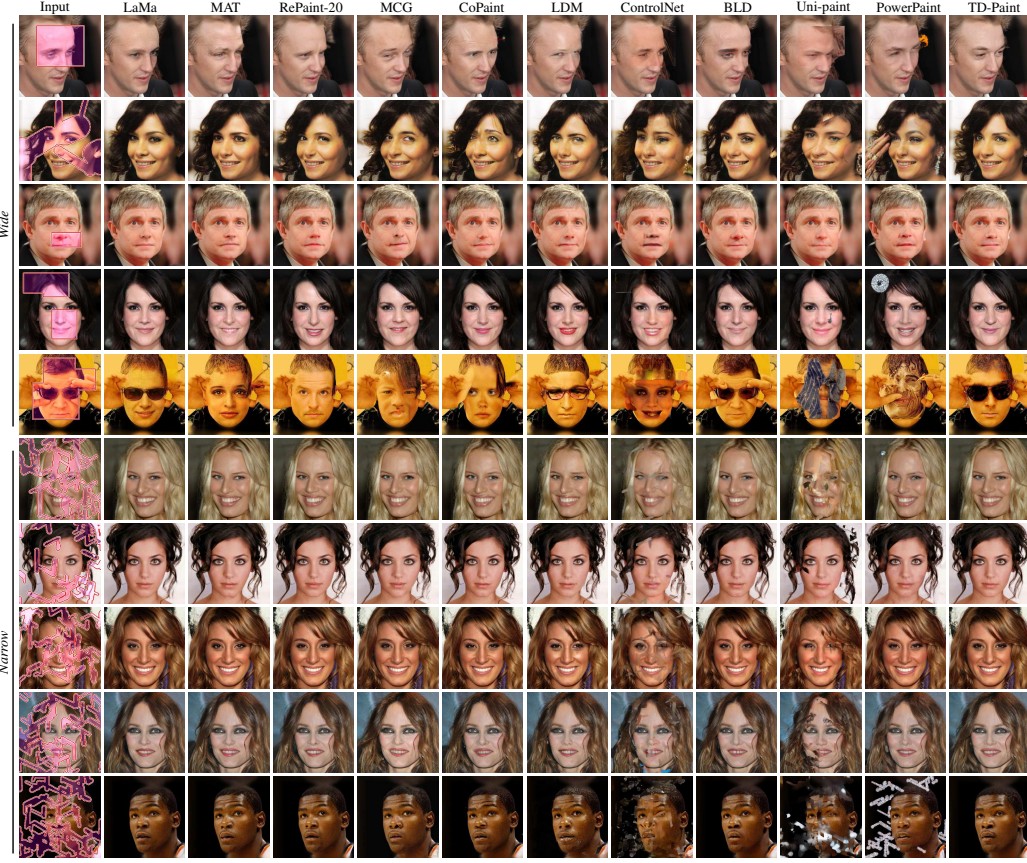

Figure 13: CelebA-HQ qualitative results

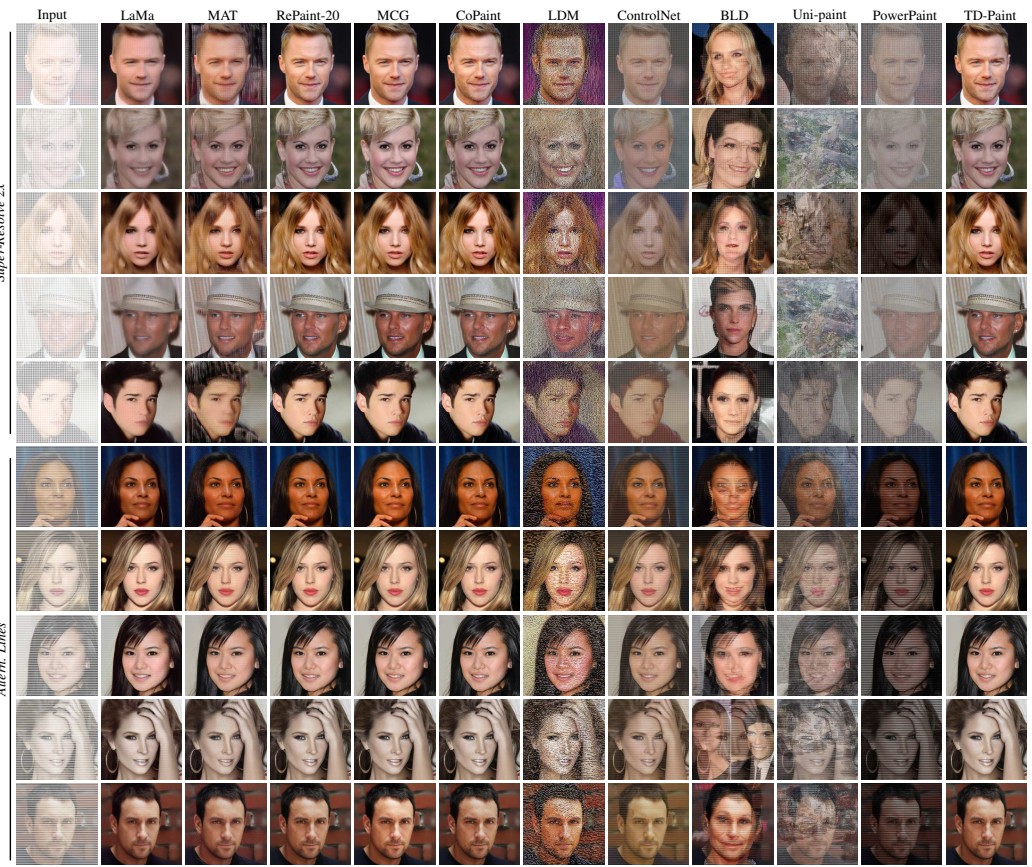

Figure 14: CelebA-HQ qualitative results

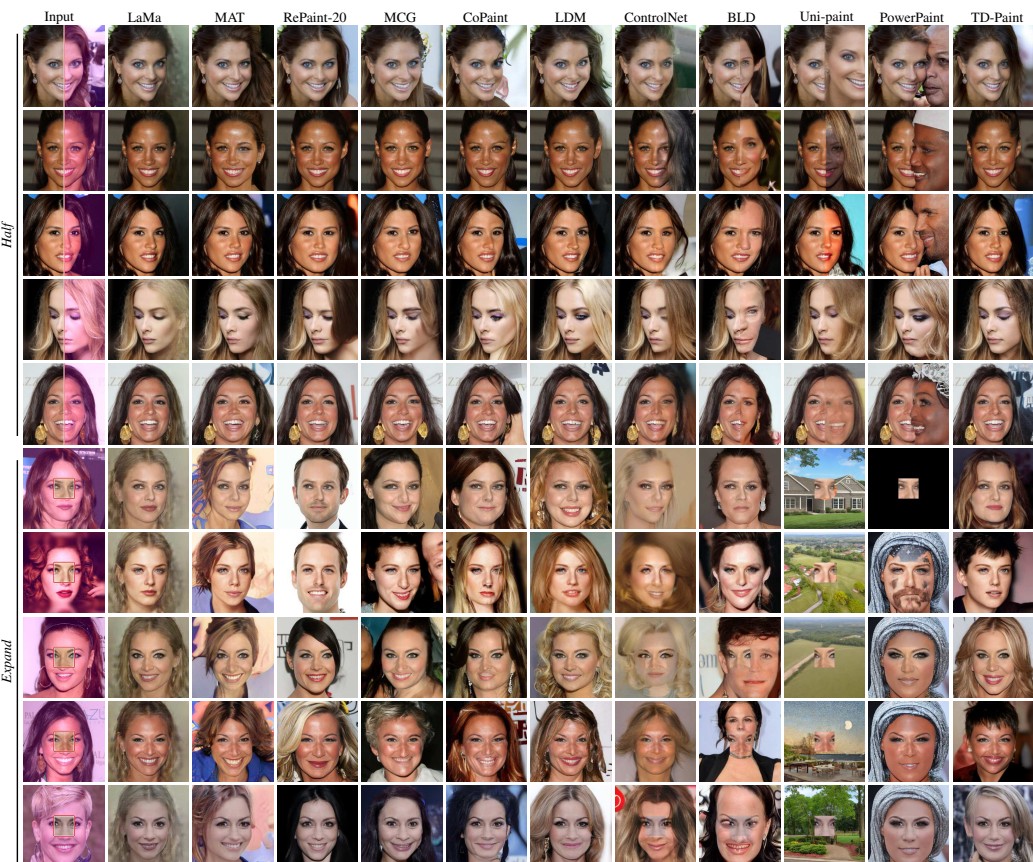

Figure 15: CelebA-HQ qualitative results

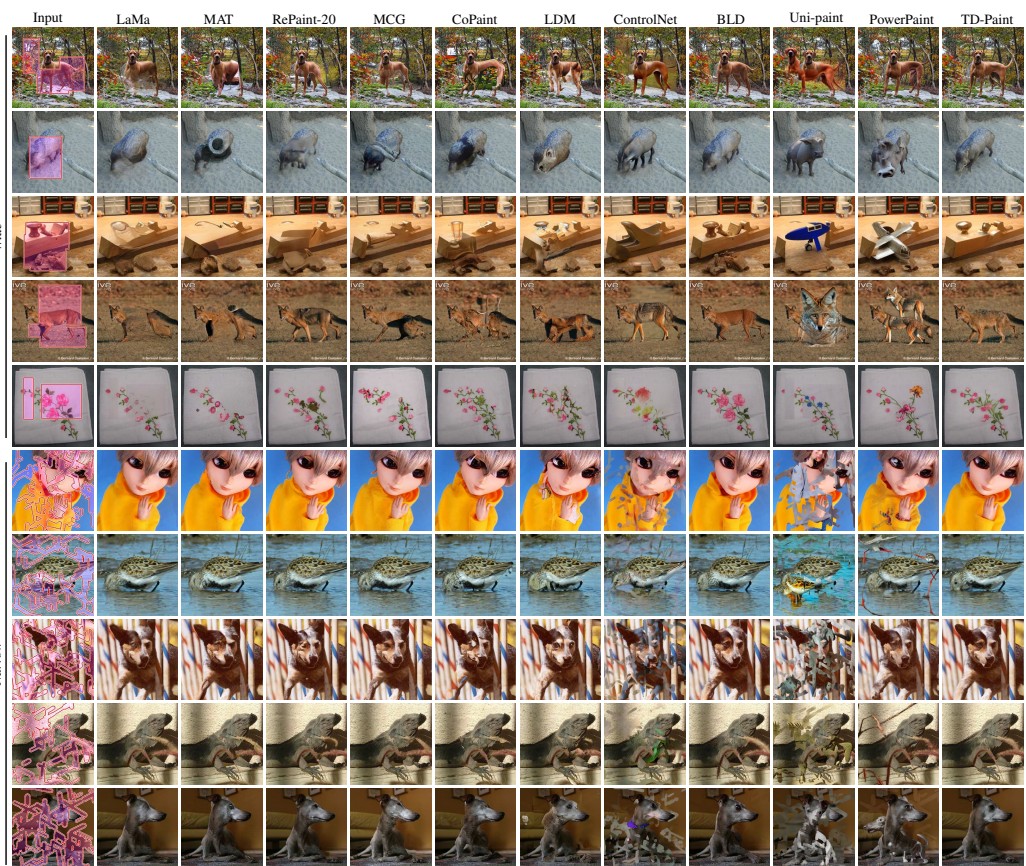

Figure 16: ImageNet1K qualitative results

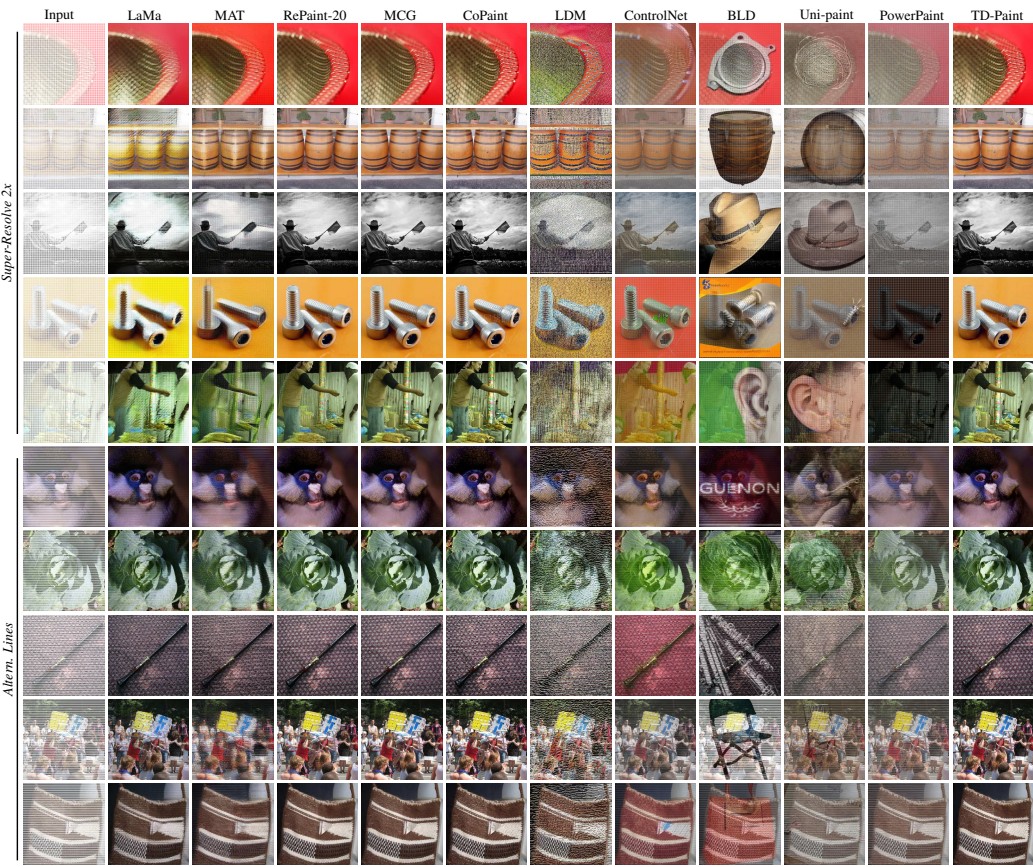

Figure 17: ImageNet1K qualitative results

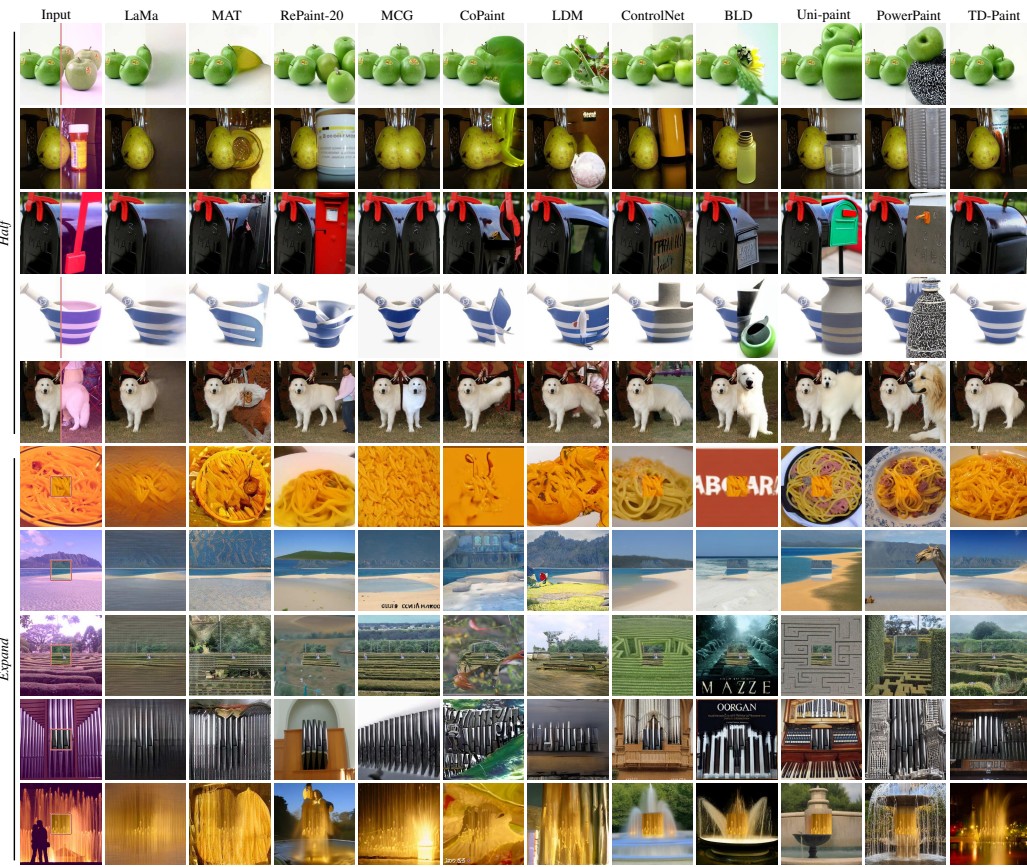

Figure 18: ImageNet1K qualitative results

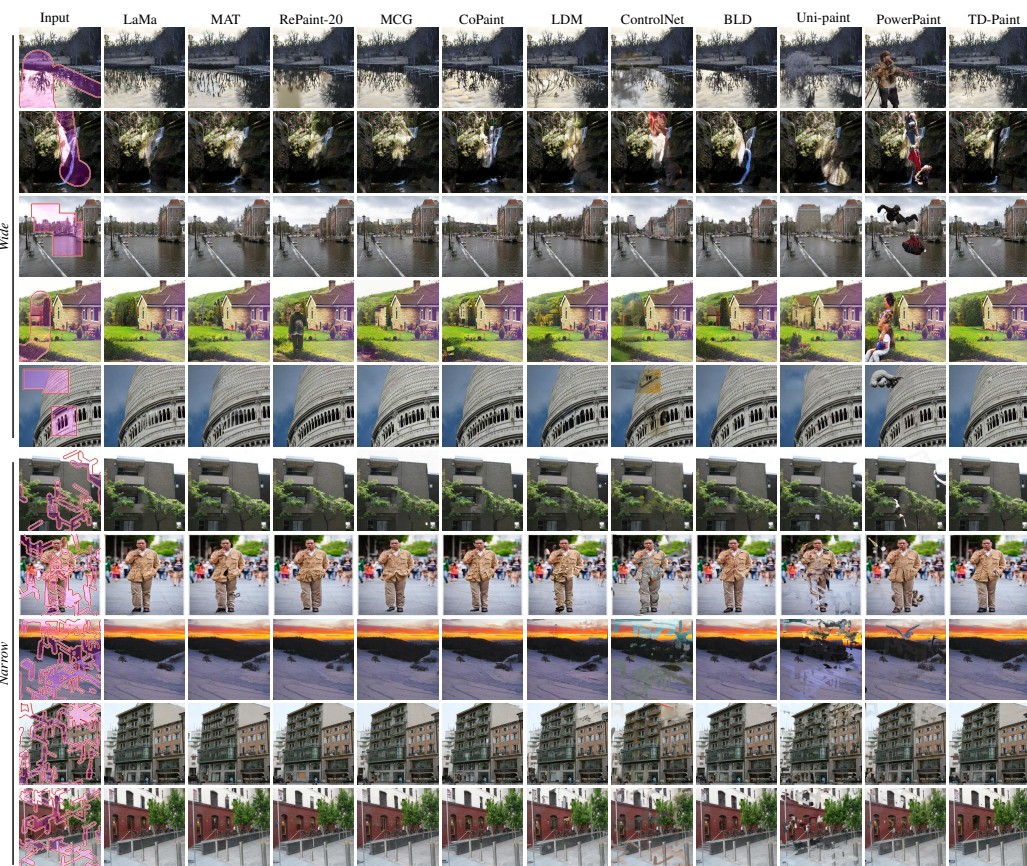

Figure 19: Places2 qualitative results

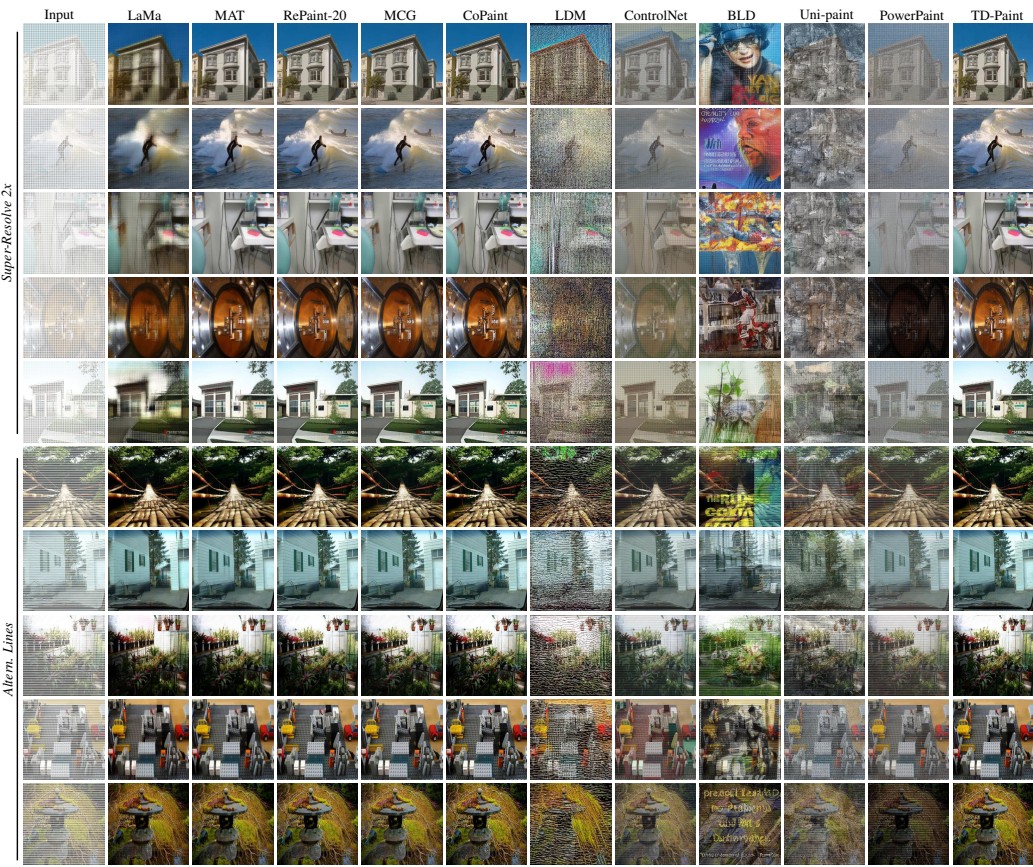

Figure 20: Places2 qualitative results

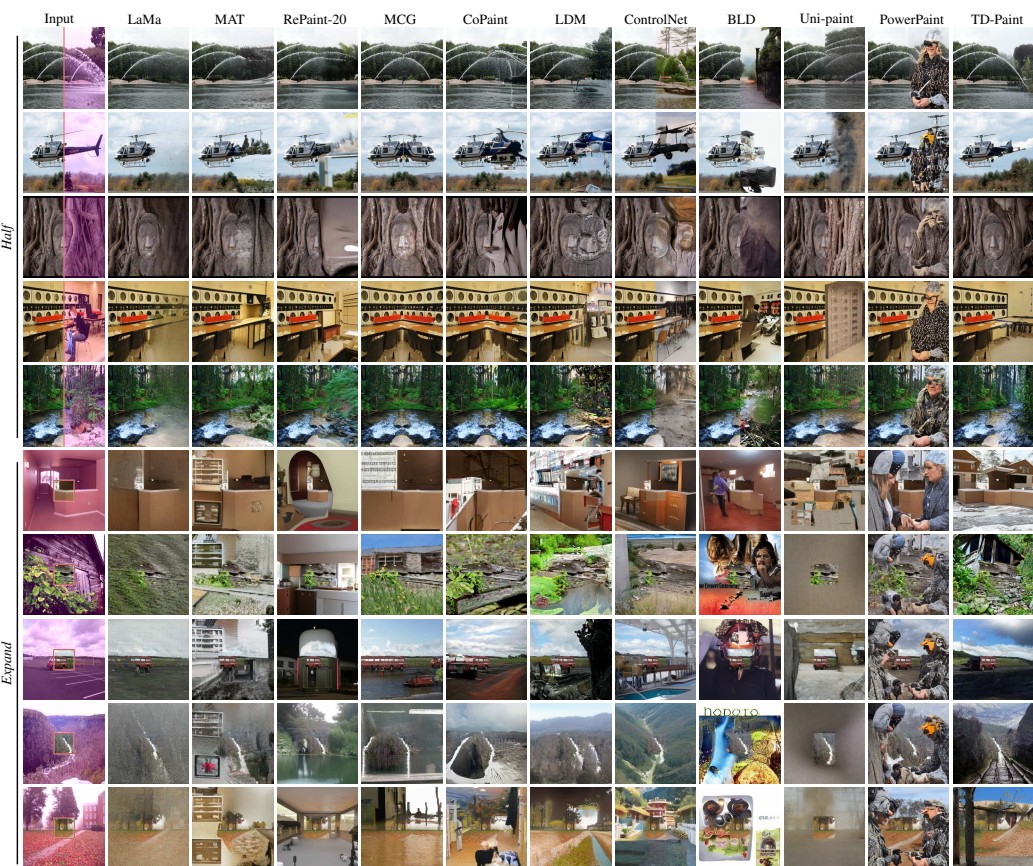

Figure 21: Places2 qualitative results

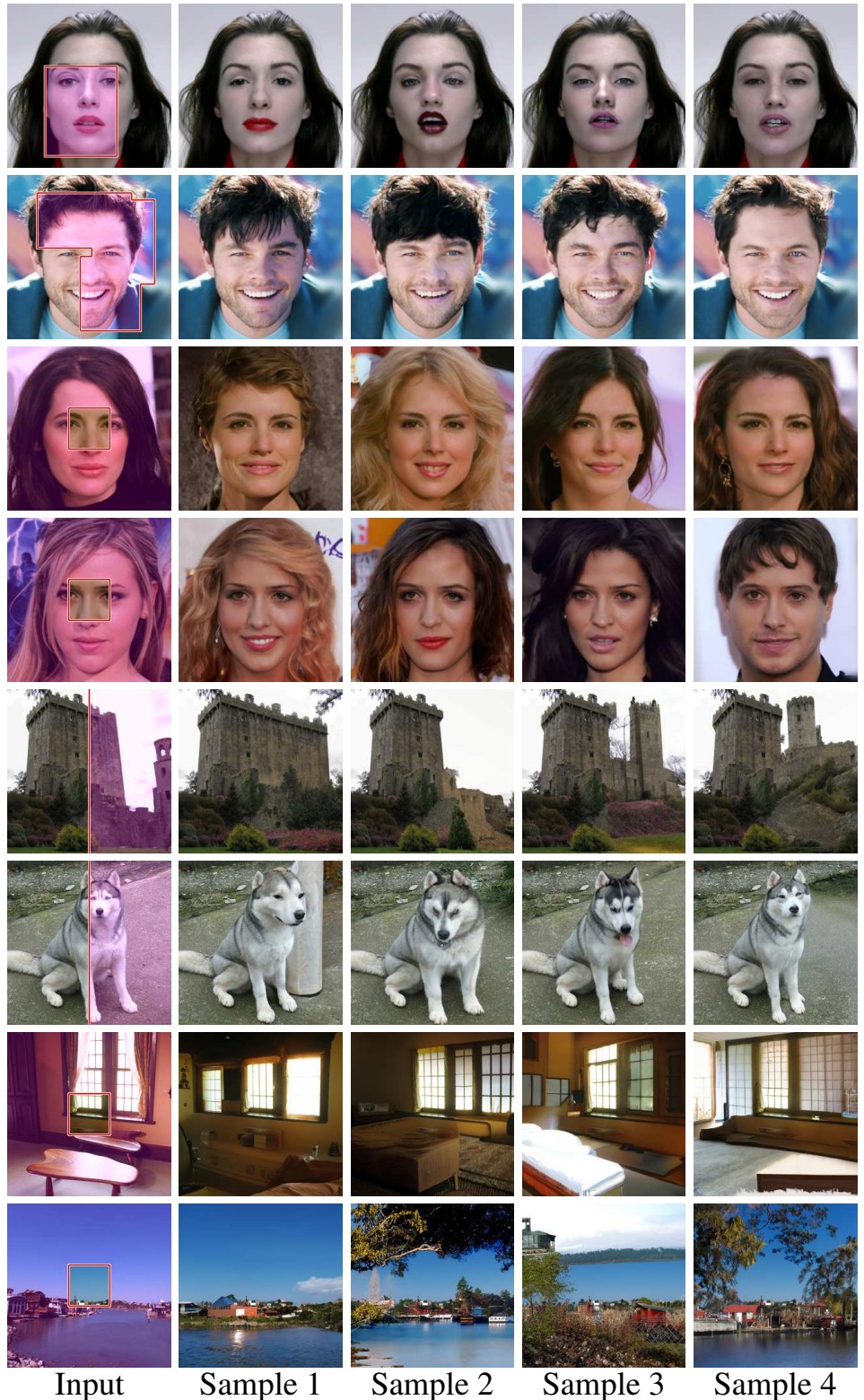

| Input | Sample 1 | Sample 2 | Sample 3 | Sample 4 |

Figure 22: Example of divers generation using TD-Paint on CelebA-HQ and ImageNet1K using the same input image and different initial noise.

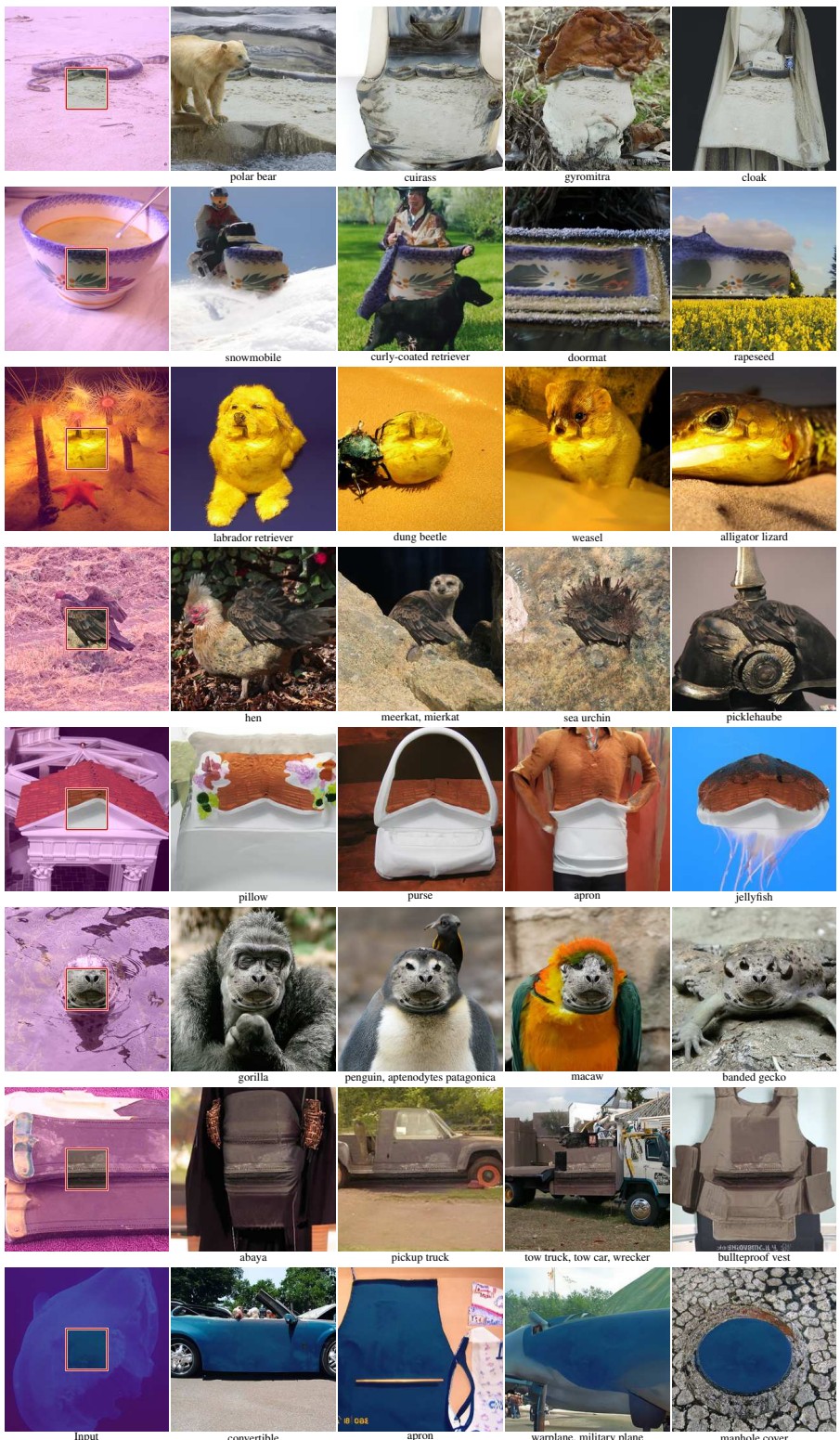

Figure 23: ImageNet1K TD-Paint diversity qualitative results using different class conditioning. For a line, TD-Paint is prompted with the same input image and mask but, with different classes.

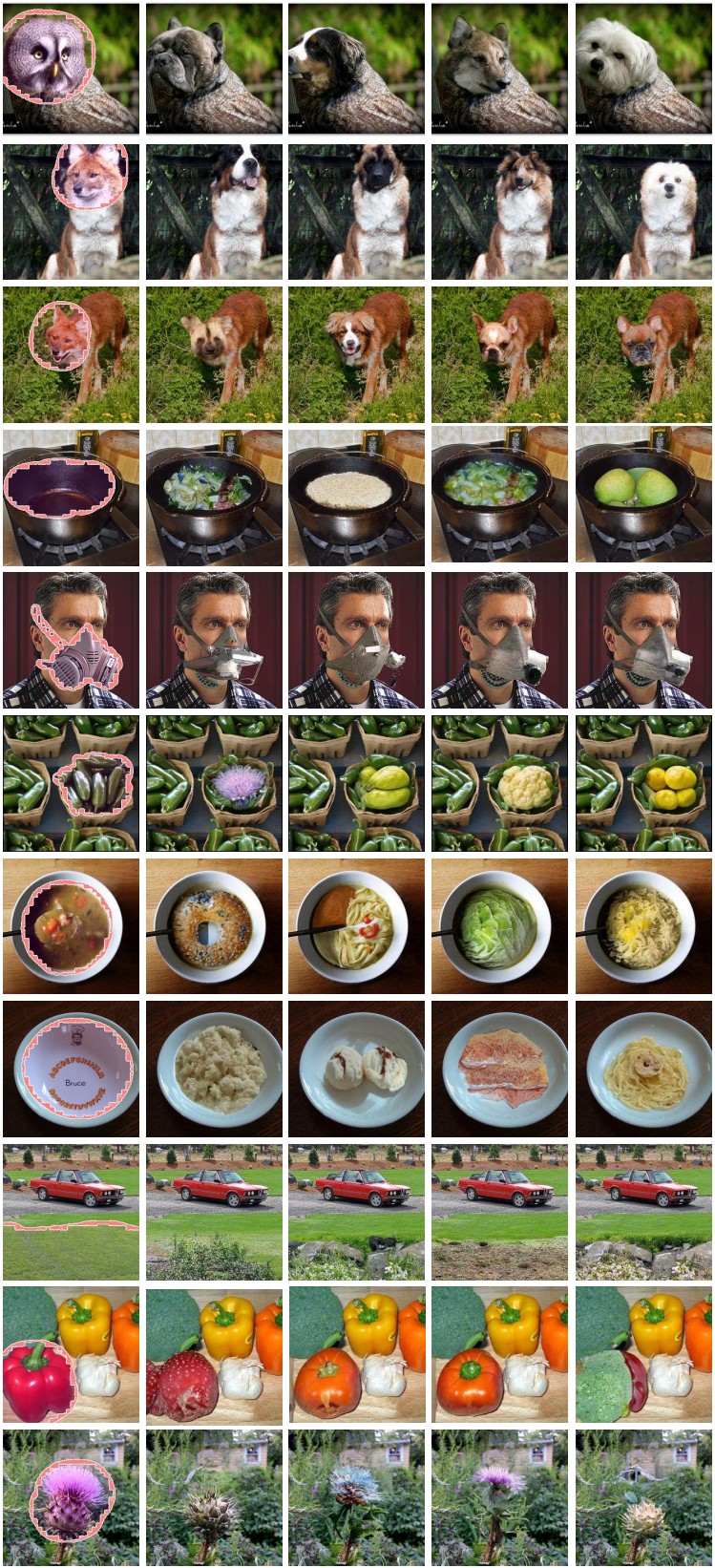

Figure 24: Demonstration of TD-paint application on the ImageNet1K dataset. The figure shows user-drawn masks highlighting specific regions or objects, followed by four generated image variations for each mask.

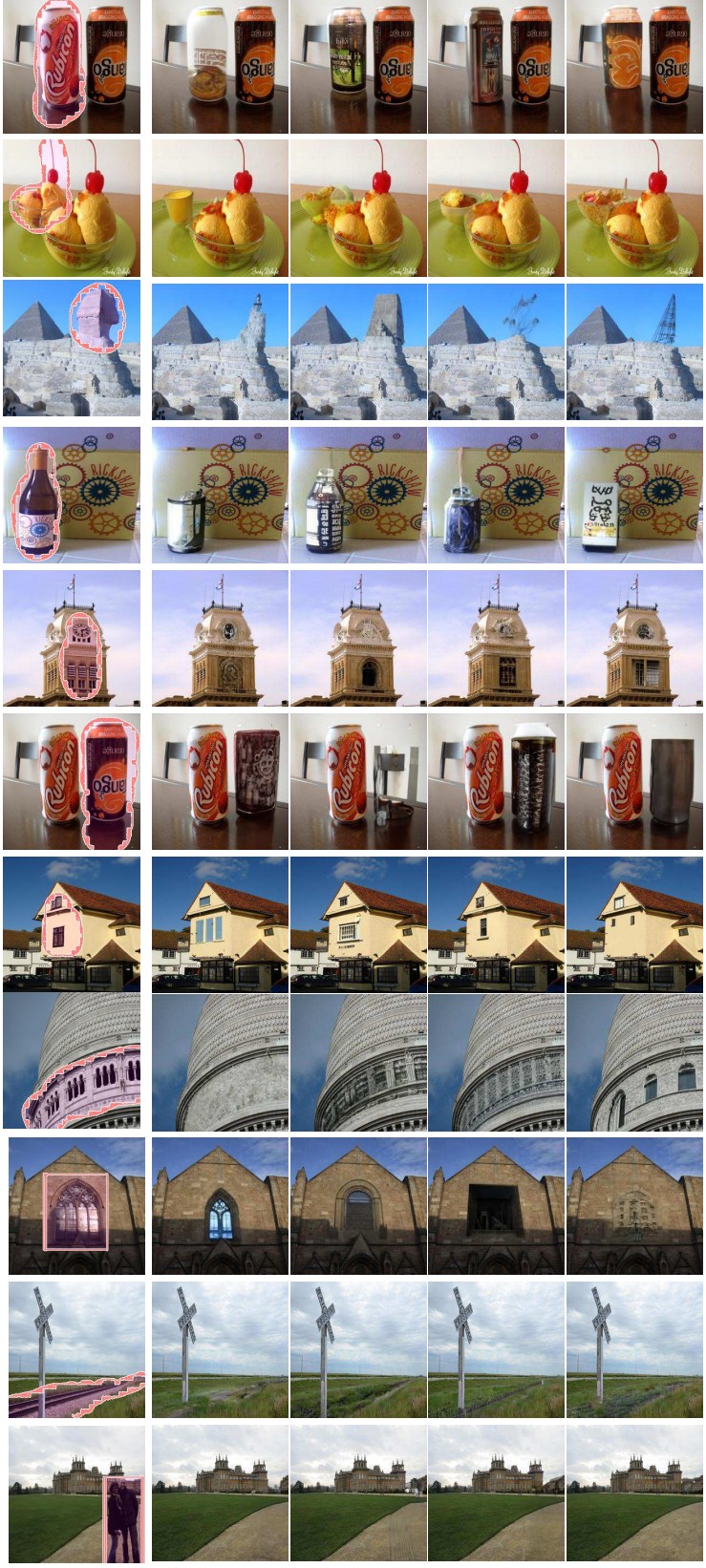

Figure 25: Demonstration of TD-paint application on the Places2 dataset. The figure shows user-drawn masks highlighting specific regions or objects, followed by four generated image variations for each mask.

