# OpenReview forum: "TD-Paint: Faster Diffusion Inpainting Through Time-Aware Pixel Conditioning"
_ICLR.cc/2025/Conference — ICLR 2025 Poster_

### Official Review · Reviewer_DEmo · 2024-10-21

**Soundness:** 3
**Presentation:** 4
**Contribution:** 3
**Rating:** 6
**Confidence:** 4

**Summary:**

The authors proposed a new training scheme of diffusion-based inpainting model that can perfectly maintain the regions inside the mask.

**Strengths:**

1. The training strategy is novel and the performance is good.

**Weaknesses:**

1. Any evaluations about how good the unchanged regions are maintained?
2. I admit that the training scheme is novel, but I am not sure if the novelty is enough for publishing. Since the entire method is a slight modification of existing diffusion model.

**Questions:**

1. The training masks are always square. How can you applied the trained model on random masks during the inference?
2. Is it possible that this method can be extended to the latent diffsusion?

---

> ### Author Response · Authors · 2024-11-24
>
> > Any evaluations about how good the unchanged regions are maintained?
>
> Since they are kept clean, their reconstruction is almost perfect. Figure 2 displays some one-step denoising examples at different noise levels. The clean regions in this figure are not replaced with the original image, and it can be observed that they are well maintained.
>
> > I admit that the training scheme is novel, but I am not sure if the novelty is enough for publishing. Since the entire method is a slight modification of existing diffusion model.
>
> While our method builds upon existing diffusion models, the time-aware approach represents a contribution that extends beyond inpainting applications. For instance, it could be adapted for privacy-preserving training by selectively applying noise to regions that require privacy protection. It could also be combined with SDEdit (as mentioned by Reviewer DSku) or used for more general approaches for accelerating diffusion inference (as mentioned by Reviewer XQUX).
>
> > The training masks are always square. How can you applied the trained model on random masks during the inference?
>
> This was indeed an important question that we had to answer. For that purpose, we provide additional training results with different types of noising strategy in Appendix A.
> We trained TD-Paint on CelebA-HQ with our patch-strategy, with generated synthetic masks (using LaMa synthetic masks), and mixing both.
> We found that while using more complicated mask patterns further boosts the model's performance, using only our patch-strategy still produces compelling results.
>
> Since the masks are defined pixel-wise, they can be applied to any random masks by setting the right pixels in the $\tau$ time map.
>
> > Is it possible that this method can be extended to the latent diffsusion?
>
> The principles of TD-Paint could potentially be adapted to latent diffusion frameworks without time-aware encoder/decoder architectures. Recent works such as Blended Latent Diffusion [Avrahami et al., SIGGRAPH 2023] and Uni-paint [Yang et al., MM 2023] have implemented operations in the latent space that share conceptual similarities with our formulation in Equation 9. However, operating in the latent space may compromise the model's ability to effectively handle fine-grained masks due to information loss during mask downsampling.
>
> It could be more relevent to use a DiT to obtain a patch-wise embedding, then apply equation 9 between known and unknown embedding.

---

> > ### Comment · Reviewer_DEmo · 2024-11-25
> >
> > I think authors solved all my concerns

---

### Official Review · Reviewer_VMkS · 2024-11-04

**Soundness:** 2
**Presentation:** 2
**Contribution:** 2
**Rating:** 6
**Confidence:** 4

**Summary:**

This paper proposes a method that can do faster inpainting in diffusion models by introducing time-aware denoising process.

**Strengths:**

1. It can generate high-quality inpainting on various mask sizes and shapes.
2. It is faster compared to other stare-of-art methods.
3. It has extensive evaluations

**Weaknesses:**

1. The visual result is not much better than other baselines
2. Even though they have many evaluation results, the improvement in image quality with scores in not much.

**Questions:**

The paper's contribution to the community is not that great, as time improvement is not that significant.

---

> ### Author Response · Authors · 2024-11-24
>
> > The visual result is not much better than other baselines
> Even though they have many evaluation results, the improvement in image quality with scores in not much.
>
> Maybe the visual results of all models appear similar for small masks. However, when evaluating the results for large masks, our model significantly outperforms the baseline models. For example, in Figures 5 and 6, none of the baseline models can generate realistic or coherent images for half and expanded masks. However, our model successfully produces realistic and coherent results across all mask types. Additionally, please refer to our supplementary results for a more detailed evaluation of the performance of our proposed model.
>
> > The paper's contribution to the community is not that great, as time improvement is not that significant.
>
> We believe that, additionally to the time improvement (more than 4x), we bring a new training paradigm that the community could further use in future works, as mentioned by Reviewer DSku (e.g., by combining it with SDEdit) and Reviewer XQUX (e.g. for a more general approach to accelerating diffusion inference).
> It could also be adapted for privacy-preserving training by selectively applying noise to regions that require privacy protection.

---

### Official Review · Reviewer_DSku · 2024-11-04

**Soundness:** 3
**Presentation:** 2
**Contribution:** 2
**Rating:** 6
**Confidence:** 5

**Summary:**

The authors mainly proposed to use the same T2I architecture to do inpainting. To properly feed in the known image conditions, they propose to use a pixel-wise noisy injection with different levels, and a spatial-variant timesteps map. Therefore, during training and inference, it allows to add minimum levels of noises to known image regions, while only diffuse the unknown regions following the normal diffusion process. The idea is simple, effective and easy to implement. Extensive experiments also demonstrate its effectiveness in maintaining good inpainting quality while reducing inpainting complexity compared to RePaint.

**Strengths:**

- The design of spatial-variant time embedding for inpainting tasks bring two major advantages: (1) no architecture change form T2I, and (2) relative faster inpainting process.
- The authors provide extensive and scientific evaluation on different masks, and the proposed method achieved good inpainting quality.

**Weaknesses:**

- Repaint seems not the state-of-the-art inpainting model, and recently models based on latest latent diffusion model or DiT framework achieve much better results, for instance, powerpaint(https://powerpaint.github.io/). Some comparison needs to be updated.
- Since the model is trained on diffusion-based framework, is it still necessary to list ConvNet-based or transformer-based model in the comparison? They will be not comparable given the inference speed and model capacity. The comparisons should focus on latest diffusion-based model in quality and speed.
- To the best of the reviewer's knowledge, recent diffusion-based inpainting is no longer using the way of RePaint to do inference. The problems mentioned by the authors like 'consistency' are no longer an issue. It can be simply achieved by adding an additional conditions concatenated with the noisy input in UNet, or adding additional tokens for DiT-based framework. Some controlnet-based method do have the composition and harmonization problems.
- It is not clear how the framework can be generalized to image editing, when the mask is soft or we need some levels of content preservation under the mask region. Without having the original image as a condition, it will be very hard to generalize the model to other editing tasks, like color changing. Therefore, the proposed framework will be limited as a foundation model, or used to finetuned for other tasks.
- The masks used in inpainting literature are mostly not suitable for real practical application because they are not used by the real inpainter users. Better to try some object-shaped masks and use those for user study.
- The proposed method may lack mask precision in a latent diffusion framework, especially for tasks like background replacement. For latent diffusion, the masks are usually applied to the pixel space, and the masked image will be encoded using the latent encoder for compression, and server as the condition. In this case, the latent diffusion model can still learn to reconstruct the high-precision mask boundary to preserve the details around the boundary. If we apply the low-res mask in the latent and apply different t in the precision of latent mask, it won't be able to leverage the advantages of the high-quality encoding.

**Questions:**

- During training, the authors use noise levels close to zero to generate noises in known region, while uses zero during inference. Any specific reasons of adding noises during training? Is it possible to simply set them as zero?
- How did the authors design the time step conditions input to the network? It is not clearly stated in the paper.
- How is the inpainting performance of the model compared with state-of-the-art commercial inpainter like Photoshop Generative Fill, MidJourney Editor, and DALL-E inpainter? No need to do detailed comparison, but a few figures / images to show the advantages or gaps will be sufficient.
- The pixel-wise time embedding is actually a region-wise time embedding? Is it possible to formulate it as only two time embeddings + one binary mask, so as to reduce the learning complexity of the model. The reviewer is not sure about the learning stability of the model. Also pixel-wise embedding can be well generalized to SDEdit, but it can be very hard to train.

---

> ### Author Response · Authors · 2024-11-25
>
> # Comparaison with PowerPaint
>
> We have incorporated comprehensive evaluations of PowerPaint in our analysis, detailed in the paper.
>
> For fine-grained masks (*Super-Resolution 2x* and *Alternating Lines*), PowerPaint, similar to other latent diffusion models, struggles to achieve effective results. This limitation stems from the loss of critical information during mask downsampling, affecting the model's ability to distinguish between conditional and generative regions.
>
> For *Wide* and *Narrow* masks, PowerPaint produces visually compelling results, though with occasional artifacts as shown in Figures 5 and 6.
>
> For *Half* masks, PowerPaint faces significant challenges and often generates unsatisfactory results, as evidenced in Figures 4, 5, and 6. Particularly on CelebAHQ, the model generally defaults to a deterministic face generation pattern (see Figure 15 in the Appendix).
>
> For *Expand* masks, PowerPaint shows several limitations:
> - On Places2, it tends to produce deterministic generations (Figure 21 in the Appendix)
> - On ImageNet1K, it struggles with seamless blending between known and unknown regions, particularly evident in the first and last rows of Figure 18 in the Appendix
> - On CelebAHQ, it defaults to deterministic mappings (Figure 15 in the Appendix)
>
> # Update on Conv and Transformer baselines
>
> We appreciate this observation. Following your suggestion, we have restructured our experimental comparisons to focus primarily on diffusion-based and latent diffusion-based models in the main paper. Therefore, we have moved the comparisons with CNN-based and transformer-based models to the appendix.
>
> # Comparaison with Latent Diffusion Models and ControlNet
>
> To address modern diffusion-based inpainting approaches that use full context during sampling, we have expanded our evaluation to include Latent Diffusion Models (LDM) and ControlNet Inpainting. Our detailed comparison with ControlNet can be found in our response to Reviewer 8fx5.
>
> While providing image and mask information in LDM yields strong results, our experiments show that TD-Paint still maintains a performance advantage. We propose that incorporating time information, as implemented in TD-Paint, could help bridge this performance gap, though this would require further empirical validation.
>
> Regarding DiT-based frameworks, we believe TD-Paint's patch-based strategy could be effectively integrated into such architectures. This integration could potentially lead to accelerated diffusion inference while maintaining TD-Paint's advantages in handling various mask configurations.
>
> # Generalization to image editing
>
> We acknowledge that TD-Paint was primarily designed and optimized for inpainting tasks. For scenarios requiring soft inpainting or partial content preservation, TD-Paint could be extended by incorporating region-specific noise levels: zero noise for preserved regions, complete noise for fully regenerated areas, and intermediate noise levels for regions where high-level details need to be maintained. This approach could be combined with methods like SDEdit to enhance refined content preservation.
>
> However, we recognize that for broader image manipulation tasks, such as global color transformations, additional research would be necessary to determine how TD-Paint could effectively complement existing approaches. This limitation indicates that there is opportunity for future work to improve and expand TD-Paint's capabilities beyond what it currently achieves.
>
> # About the mask shape
>
> TD-Paint's architecture is designed to be mask-shape agnostic and not specifically made for image-editing. Our extensive experiments demonstrate effectiveness across a diverse range of mask shapes and sizes, from fine-grained details to large regions.
>
> For practical applications, we provide evidence in Figure 22 (Appendix) showing TD-Paint's capability in real-world editing scenarios, such as modifying facial features (lips) and changing hairstyles. These examples demonstrate TD-Paint's potential for practical applications in image editing software.
>
>
> # Mask precision in the latent space
>
> Our choice to apply the mask in pixel-space was indeed motivated by these considerations. However, it is important to acknowledge that recent works, such as LatentPaint, Blended Latent Diffusion, and Differential Diffusion, have demonstrated success in implementing inpainting masks in low-resolution spaces. These approaches effectively combine conditional latent representations with generated latents. Following similar principles, TD-Paint could potentially be adapted to encode the masked condition and merge it with the generated latent according to the inpainting mask, aligning with established practices in latent diffusion inpainting methods.

---

> ### Author Response · Authors · 2024-11-25
>
> # Training noise level
>
> We would like to clarify that in our implementation, we maintain completely noise-free (zero noise) known regions during both training and inference phases. Furthermore, we do not compute the loss on the known regions during training. This design choice was motivated by several factors:
>
> 1. Training-inference consistency: This approach ensures identical behavior between training and inference phases
> 2. Information preservation: Zero noise in known regions maximizes the fidelity of conditional information available to the model
> 3. Learning efficiency: This simplification allows the model to focus exclusively on learning the inpainting task
>
> Our experimental results confirm that this approach leads to stable training in TD-Paint. While we explored computing loss on known regions and adding noise during training, we found no compelling advantages to these alternatives.
>
> # Time step conditioning
>
> The time-step conditioning architecture is fully detailed in Section 4.2. Let us clarify both the classical approach and our extension:
>
> The time-step conditioning architecture is fully detailed in Section 4.2. Let us clarify both the classical approach and our extension:
>
> In classical UNet-based diffusion models:
> 1. A scalar time value $t$ is embedded through a time-embedding operation: $\gamma = time embedding(t) \in R^{tdim}$
> 2. At each UNet level $l$, an MLP transforms $\gamma$ to match the current channel dimension: $\gamma' = MLP_l(\gamma) \in R^{c_l}$
> 3. $\gamma'$ is then replicated across spatial dimensions and applied dimension-wise (equation 12), resulting in shape $[h_l, w_l, c_l]$.
>
> In TD-Paint, we extend this to pixel-wise time conditioning:
> 1. Instead of a single time value, we use a time-map $\tau$ with different times per pixel
> 2. This time-map undergoes time-embedding to produce $\Gamma \in R^{[h,w,tdim]}$ (equation 13)
> 3. At each UNet level, $\Gamma$ is downscaled using operator $D$ to match spatial dimensions, yielding $\Gamma' \in R^{[h_l,w_l,c_l]}$ (equation 14)
> 4. The resulting $\Gamma'$ is applied similarly to the classical approach as it is of the same shape.
>
> For efficiency, when $D$ is implemented as min or max-pooling, operations can be optimized to represent the time-map $\tau$ using only two scalar values and a binary map.
>
> # Comparaison with commercial tools
>
> While we have conducted extensive comparisons with academic state-of-the-art models, we have not yet performed comprehensive evaluations against commercial solutions like Photoshop Generative Fill, MidJourney Editor, and DALL-E's inpainting tool. We acknowledge that such comparisons would provide valuable insights into TD-Paint's practical utility.
>
> # Pixel-wise or region-wise time embedding
>
> Our approach implements true pixel-wise time embedding, as each pixel can be theoretically be assigned a distinct time value. In practice, TD-Paint uses two distinct time values during training: one for the masked region and another for the known region. While this implementation could be viewed as "region-wise" in terms of time assignment, the underlying architecture maintains pixel-level granularity in how these time values are processed and embedded.
>
> The model can indeed be optimized using only two scalar time values (t and 0) when combined with min or max pooling as the downscaling operator D, as described in Section 4.2. While we did not explore this optimization in our current experiments, it presents a promising direction for improving computational efficiency.
>
> Regarding training stability, our model demonstrates robust convergence characteristics and achieves superior results compared to Latent Diffusion Inpainting and ControlNet under similar training conditions.
>
> The pixel-wise embedding framework we propose is inherently modular and could be integrated with other diffusion-based methods, including SDEdit. This extensibility suggests promising research directions for combining our time embedding mechanism with existing techniques to further advance the state-of-the-art in image inpainting.

---

> > ### Author Response · Authors · 2024-11-26
> >
> > > The masks used in inpainting literature are mostly not suitable for real practical application because they are not used by the real inpainter users. Better to try some object-shaped masks and use those for user study.
> >
> > To demonstrate TD-Paint's capabilities in practical inpainting scenarios, we provide comprehensive examples across multiple datasets. Figure 22 (Appendix) illustrates realistic user inpainting masks on the CelebAHQ dataset, while Figures 24 and 25 present additional new examples using the ImageNet1K (class-conditonned) and Places2 (without condition) datasets, respectively.
> >
> > These examples feature user-drawn masks that naturally follow object boundaries and regional structures, providing a realistic representation of how users interact with inpainting tools. The results demonstrate TD-Paint's robust performance in practical image manipulation tasks. Notably, TD-Paint successfully generates coherent images while maintaining class-conditional consistency, even in cases where the specified class label differs from the original image semantics (as shown in the first row of Figure 24).
> > Furthermore, the model effectively fills regions and replaces objects while maintaining scene consistency.

---

> ### Comment · Reviewer_DSku · 2024-11-27
>
> Thanks for all the detailed explanations and revisions of the paper.
> - I am not sure whether it is fair to compare on some masks on which the baseline models were not trained. For example, SR masks or line masks are not typical way an inpainting model was trained on. So comparing on them may not be the best way to claim the model advantages.
> - Maybe it's better to clarify, given the current LDM-based or controlnet-based methods, why the proposed method is still "faster" or "more stable". And when we claim it is faster, do we have strong evidences on that?
> - The paper claims "pixel-wise" time embedding, while in practice the training is only conducted on two values. I feel it is still misleading and there is not enough evidence to show that true pixel-wise (each pixel has a different values) time embedding does not have training convergence issues or other issues. It makes the paper not very convincing and a bit over-claimed.

---

> ### Author Response · Authors · 2024-11-27
>
> ### Remark 1
> We appreciate this important point about mask comparison fairness. To begin, we would like to clarify that TD-Paint does not use the SR mask or the line mask during training.
> Below, we outline our evaluation methodology across different baseline models:
> 1. **Zero-shot Models** (RePaint, MCG, CoPaint, Blended Latent Diffusion)
>    - These models are designed to operate without training on specific mask types
>    - Evaluating their performance on SR and line masks is methodologically appropriate, as the models claim to be applicable to various mask configurations
>
> 2. **Few-Shot Models** (Uni-Paint)
>    - Uni-Paint uses few-shot learning by fine-tuning on given image-mask pairs
>    - Our comparison is valid since this approach rely on pretrained models and perform optimization using the test image and mask information
>
> 3. **Synthetic Mask-Trained Models** (LaMa, MAT, LDM, ControlNet)
>    - All models are trained using LaMa-style synthetic masks or closely related strategies
>    - Our comparison is appropriate as we employ similar training masks
>    - In response to the concern about performance consistency, our ablation A studies confirm that TD-Paint exhibit minimal variation in performance with regard to mask strategy, even on challenging SR and line masks, supporting the robustness of our model
>
> 4. **PowerPaint-mask**
>    - Uses masks derived from object bounding boxes, segmentation, and random masks
>    - This approach aligns with ours, as the random mask selection produces results comparable to those obtained with large masks in LaMa/MAT.
>
> Our empirical evidence, presented in Appendix A, demonstrates TD-Paint's robust performance on SR and line masks, despite not being explicitly trained on these mask types. When trained exclusively on LaMa masks, TD-Paint:
> - Achieves comparable (or better) results to leading zero-shot models (RePaint-20, MCG, CoPaint)
> - Outperforms models trained on identical/similar masks (LaMa, MAT, LDM, ControlNet)
> - Demonstrates superior performance compared to models with mask-specific fine-tuning capabilities (Uni-Paint)
>
> Furthermore, TD-Paint achieves superior performance on Half and Expand masks, as evidenced by the KID metrics in Table 2, despite the absence of explicit training on these mask types. These results highlight TD-Paint's strong generalization capabilities across a wide range of inpainting scenarios, from thin to very large masks.
>
> ### Remark 2
>
> We acknowledge the concerns regarding stability when using our patch-based strategy. As shown in Ablation Study A, our experiments combining LaMa masks with the patch-based approach indicate that both strategies deliver robust results.
> To provide additional evidence of stability, we provide new epoch-wise results for both *wide* and *narrow* masks. The LPIPS metrics show stability throughout training, with TD-Paint achieving even better LPIPS scores than those reported in the paper for complex datasets such as ImageNet1K and Places2.
> Specifically, epochs 20 to 22 show improved performance on ImageNet1k, and epoch 24 demonstrates better results on Places2.
>
> Table: TD-Paint LPIPS↓ vs Training Epochs on CelebAHQ dataset
> ||149|249|349|449 (used)|499|
> |-|-|-|-|-|-|
> |Wide|0.0599|0.0579|0.0564|0.0554|0.0556|
> |Narrow|0.0308|0.0287|0.0281|0.0275|0.0276|
>
> Table: TD-Paint LPIPS↓ vs Training Epochs on ImageNet1K dataset
> ||10|12|14|16|18|19 (used)|20|21|22|
> |-|-|-|-|-|-|-|-|-|-|
> |Wide|0.1030|0.1009|0.0998|0.0996|0.1053|0.0995|0.0995|0.0988|0.0991|
> |Narrow|0.0604|0.0585|0.0575|0.0575|0.0592|0.0573|0.0571|0.0571|0.0567|
>
> Table: TD-Paint LPIPS↓ vs Training Epochs on Places2 dataset
> ||16|17|18 (used)|19|20|21|22|23|24|
> |-|-|-|-|-|-|-|-|-|-|
> |Wide|0.1435|0.1133|0.1116|0.1139|0.1113|0.1150|0.1106|0.1132|0.1107|
> |Narrow|0.0799|0.0658|0.0640|0.0640|0.0626|0.0636|0.0623|0.0633|0.0609|
>
>
> Regarding inpainting speed, our study evaluates various types of models:
> - CNN-based approaches (LaMA)
> - Transformer-based models (MAT)
> - Pixel-space diffusion models (RePaint, MCG, CoPaint)
> - Latent-space diffusion models (added in this rebuttal: LDM inpaint, ControlNet inpaint, Blended Latent Diffusion, Uni-Paint and PowerPaint)
>
> Latent diffusion models operate in a semantically compressed space, while pixel-space diffusion models work directly in the high-resolution pixel space. Our speed improvements primarily focus on comparisons with pixel-space diffusion models. For a comprehensive and fair comparison with latent-space methods, TD-Paint would need to be reimplemented with a latent diffusion architecture. We consider this a promising direction for future work, where TD-Paint's speed could be evaluated against similar latent-space baselines.

---

> ### Author Response · Authors · 2024-11-27
>
> ###  Remark 3
>
> The term "pixel-wise" refers to our model's capability to assign independent time values to each pixel, regardless of neighboring pixels' values. While our experiments primarily use two distinct time values, the time map $\tau$ preserves the full spatial resolution of [height $\times$ width], allowing each pixel in the image to be assigned a time value $t$.
> This pixel-level granularity in time embedding is a key factor contributing to TD-Paint's robust performance on challenging mask configurations. By embedding time at the pixel level rather than globally or in blocks, the architecture enables fine-grained control over conditions at each spatial location, enhancing its adaptability and precision.
>
> Furthermore, TD-Paint can effectively be seen as using more than two $t$ values. This is because the time map $\tau$ is downscaled with bilinear interpolation (see Eq. (13)) *prior* to time embedding, resulting in a continuous range of $t$ values across the spatial dimensions at each UNet resolution.
>
> We acknowledge that implementing completely arbitraty time values for each pixel (i.e. in the "worst" case as much different $t$ as pixel-coordinate, |{t | $\forall$ t $\in$ $\tau$}| = h $\times$ w) would likely require careful consideration of the training procedure.
>
> However, in the context of inpainting tasks as described in our paper, it remains unclear whether such an approach would provide additional benefits beyond our current implementation.
>
> It is worth noting that multiple time values during training has proven effective in various contexts, such as in Diffusion Forcing ["Diffusion Forcing: Next-token Prediction Meets Full-Sequence Diffusion" NeurIPS 2025], where varying time values were effectively applied at the token level for sequence generation tasks.

---

> > ### Comment · Reviewer_DSku · 2024-11-30
> >
> > Thank you for the detailed response! I believe the term "pixel-wise map" may be somewhat overstated. It gives the impression that the proposed method is a generic framework capable of training each pixel independently with different timestep. While the authors have indeed proposed such a framework and recent works apply similar ideas to tokens, there is no direct experimentation to substantiate this claim. The continuous values resulting from bilinear downsampling do not seem sufficient to fully validate the framework. Perhaps it would be worth tempering this claim slightly.
> >
> > Additionally, I see great potential for the inpainting task with a soft mask beyond its current implementation. This could be an exciting direction for the authors to explore in future work.

---

> > > ### Author Response · Authors · 2024-11-30
> > >
> > > >Thank you for the detailed response! I believe the term "pixel-wise map" may be somewhat overstated. It gives the impression that the proposed method is a generic framework capable of training each pixel independently with different timestep. While the authors have indeed proposed such a framework and recent works apply similar ideas to tokens, there is no direct experimentation to substantiate this claim. The continuous values resulting from bilinear downsampling do not seem sufficient to fully validate the framework. Perhaps it would be worth tempering this claim slightly.
> > >
> > > We acknowledge that the term "pixel-wise map" may have conveyed the impression that the noise would be completely independent for each pixel.
> > > The use of this terminology was intended to describe the spatial resolution of the time-map at the pixel level, rather than implying the capability to train each pixel independently with different timesteps.
> > > In the context of inpainting, we believed such granular control was not necessary, however, we will adjust this claim accordingly in the final version of the manuscript, by defining the distinct timesteps: one associated with the known part $x^\oplus$ ($t^\oplus=0$) and another with the unknwon part $x^\ominus$ ($t^\ominus=t$), as detailed in the generation procedure of our method discussed in line 251.
> > >
> > > > Additionally, I see great potential for the inpainting task with a soft mask beyond its current implementation. This could be an exciting direction for the authors to explore in future work.
> > >
> > > Regarding future directions, we agree that extending the framework to incorporate soft masks, as well as potential implementations using Latent Diffusion Models (LDM) or DiT architectures would be promising areas of research. However, we emphasize that the current version of TD-Paint demonstrates strong performance, particularly in terms of KID metrics, outperforming other state-of-the-art methods in both pixel and latent spaces.
> > > Indeed, our decision to not directly study LDM models was motivated by their observed limitations with very thin masks, especially in super-resolution and alternating lines.
> > >
> > > We sincerely appreciate your constructive comments and hope that our responses address your concerns and justify an increase in your overall rating of our manuscript.

---

> > > > ### Comment · Reviewer_DSku · 2024-12-02
> > > >
> > > > Thanks for the detailed and patient explanations.
> > > > I encourage the authors to tune down the claim a bit, so the pixel-wise map is not that confusing. I will increase my score a bit.

---

> > > > > ### Author Response · Authors · 2024-12-02
> > > > >
> > > > > We appreciate the reviewer's constructive feedback. The claims regarding the pixel-wise map will be revised as detailed in our previous response and included in the final version of the manuscript.

---

### Official Review · Reviewer_8fx5 · 2024-11-05

**Soundness:** 3
**Presentation:** 3
**Contribution:** 3
**Rating:** 6
**Confidence:** 4

**Summary:**

The paper presents a novel method for speeding up diffusion-based inpainting. It changes the noised known region condition in the RePaint paper into a clean one for condition. To support this change, it introduces a trained diffusion model that supports a spatial-varying timestep condition to mark the known and missing part of the image. This new design also allows the method to avoid the repeated sampling process in RePaint, greatly increasing the speed. Quantitative results also demonstrate the proposed method performs on par or better with existing methods.

**Strengths:**

1. Training the diffusion model to directly use clean known parts as the diffusion condition, avoiding complex modifications to the diffusion process. This approach significantly speeds up the method compared to existing approaches, enhancing efficiency without compromising quality.
2. Introduction of a novel spatial-varying timestep condition for injecting the known region mask. This innovative technique allows for more precise control over the inpainting process, resulting in improved output quality.
3. Comprehensive testing on multiple diverse datasets, demonstrating superior performance (for both quality and diversity) across various scenarios. The extensive evaluation provides strong evidence for the method's robustness and effectiveness in different contexts.

**Weaknesses:**

The paper is in general good, I only have concerns on its generality:

Since the proposed method requires training and is not trained with data besides CelebA-HQ, ImageNet1K, and Places2, it probably cannot generalize to open-world images like LaMa.

**Questions:**

1. Training Inference Gap: The training is done on patch-wise known regions. But inference of the known region doesn’t necessarily be axis-aligned patches. How do you solve this gap? Will training on randomly shaped known regions further boost the model’s performance?
2. I wonder how well the proposed method performs when adapted to larger diffusion models like stable diffusion.
3. The ControlNet-Inpaint model also provides fast and high-quality diffusion-based inpainting. How does the proposed method perform compared to that model?

I would raise my score if these questions were answered.

---

> ### Author Response · Authors · 2024-11-25
>
> > The paper is in general good, I only have concerns on its generality:
> Since the proposed method requires training and is not trained with data besides CelebA-HQ, ImageNet1K, and Places2, it probably cannot generalize to open-world images like LaMa.
>
> This limitation is not inherent to TD-Paint's architecture or methodology, but rather reflects the scope of the training datasets we utilized in our current experiments. The generalization capabilities of TD-Paint could be further demonstrated through training on larger-scale text-image pair datasets, although such experiments would require significant computational resources.
> Importantly, there is no theoretical indication that TD-Paint would not generalize effectively to datasets even larger than Places2.
>
> > Training Inference Gap: The training is done on patch-wise known regions. But inference of the known region doesn’t necessarily be axis-aligned patches. How do you solve this gap? Will training on randomly shaped known regions further boost the model’s performance?
>
> To directly address this important about the training-inference gap, we conducted comprehensive experiments with different noising strategies, documented in Appendix A. We evaluated three training configurations on CelebA-HQ:
> 1. Our standard patch-strategy
> 2. LaMa's synthetic mask generation
> 3. A combination of both approaches
>
> While our results show that incorporating more complex mask patterns (like LaMa's synthetic masks) does enhance model performance, the patch-strategy alone achieves strong results. This demonstrates that our base approach generalizes well to non-aligned masks during inference, while suggesting potential performance gains through diverse mask patterns during training.
>
> > I wonder how well the proposed method performs when adapted to larger diffusion models like stable diffusion.
>
> Our experimental results demonstrate TD-Paint's scalability across datasets of increasing complexity: from the domain-specific CelebAHQ to the more diverse ImageNet1K, and finally to the large-scale Places2 dataset. The consistent performance across these datasets suggests that TD-Paint's core principles could be effectively adapted to larger architectures like Stable Diffusion. While such an implementation would require additional engineering efforts and computational resources, there are no fundamental architectural limitations that would prevent such scaling. However, empirical validation would be necessary to confirm this hypothesis and optimize the implementation for larger-scale models.
>
> > The ControlNet-Inpaint model also provides fast and high-quality diffusion-based inpainting. How does the proposed method perform compared to that model?
>
> To address this comparison, we conducted additional experiments with Latent Diffusion Inpainting (LDM) [1] and ControlNet-Inpaint (CN) [2], both of which utilize the whole image as conditional context.
>
> For *Wide* and *Narrow* masks, both LDM and CN produce more artifacts than TD-Paint, as evidenced by the KID metrics and visual results in Figures 4, 5, and 6. This effect is more pronounced with CN, while LDM approaches TD-Paint's performance for these mask configurations.
>
> For fine-grained masks (*Super-Resolution 2x* and *Alternating Lines*), both latent diffusion models fail to solve the tasks effectively. This limitation stems from the necessary downsampling of inpainting masks, which results in significant information loss and for the latent space combination between known and unknown regions.
>
> For very large masks (*Half* and *Expand*) on more complex datasets (ImageNet1K and Places2), LDM produces compelling results, though not matching TD-Paint's performance, as demonstrated by the KID metrics in Table 2 and qualitative results in Figures 18 and 21. We hypothesize that incorporating time-awareness into latent diffusion models could enhance their inpainting capabilities and potentially bridge the performance gap between TD-Paint and LDM-based methods.
>
>
> [1] "High-resolution image synthesis with latent diffusion models." CVPR 2022
>
> [2] "Adding conditional control to text-to-image diffusion models." ICCV 2023

---

> > ### Comment · Reviewer_8fx5 · 2024-11-26
> >
> > Thanks for the clarifications for the questions. I think the paper would be stronger with explorations conducted on larger diffusion models and open-world data. I will remain my current positive score.

---

### Official Review · Reviewer_XQUX · 2024-11-09

**Soundness:** 3
**Presentation:** 3
**Contribution:** 2
**Rating:** 6
**Confidence:** 4

**Summary:**

This paper proposes a way to perform inpainting by leveraging pre-trained pixel diffusion model. The authors fine-tune a diffusion model such that instead of inputting a single scalar that represents the tilmestep of the noise in the diffusion process, the model inputs a map, representing per-pixel tilmesteps. Then, at inference, this allows them to always input the fully denoised image in the known regions while iteratively denoising the masked areas. Unlike the main baseline, RePaint, this process is much faster, as it does not require multiple diffusion passes to synchronize the known and unknown regions.

**Strengths:**

The proposed approach is quite simple but at the same time intuitive and effective. The idea of varying the diffusion at a pixel level is a promising one, both for novel applications (such as inpainting) as well as for a more general approach to accelerating diffusion inference.

**Weaknesses:**

My main concern has to do with lack of comparison to or discussion of some more recent and relevant SOTA diffusion-based inpainting approaches. For instance, LatentPaint [Corneanu et al., WACV 2024] also a propose an approach that requires fine-tuning with a fast inference pass (compared to RePaint). Additionally, "Blended Latent Diffusion" [Avrahami et al., SIGGRAPH 2023] and "Uni-paint" [Yang et al., MM 2023] leverage latent diffusion models for an inpainting task (while the former does not explicitly target unconditional inapinting, it ca be used for that purpose).  It would be very helpful to see comparisons to some of these recent works as well as a high-level discussion of their differences in order to evaluate the proposed method.

On a more minor note, the presentation and exposition could be improved. Figure 1, which illustrates the proposed approach in contrast to RePaint, is somewhat difficult to follow and includes notation that is not defined until much later. In contrast, the idea behind RePaint is explained multiple times in various sections throughout the paper and could be truncated.

I would also encourage the authors to cite and discuss "Differential Diffusion: Giving Each Pixel Its Strength" [Levin and Fried, 2023]. Given that this work does not appear in a peer-reviewed publication, I am not considering it in my rating, but given its close relevance to the topic, it would be good to mention.

**Questions:**

How can the proposed method be used to scale up to higher resolution images? For instance, is there a straightforward way to extend to LDMs, as some of the more recent inpainting approaches have been doing?

---

> ### Author Response · Authors · 2024-11-25
>
> > My main concern has to do with lack of comparison to or discussion of some more recent and relevant SOTA diffusion-based inpainting approaches. For instance, LatentPaint [Corneanu et al., WACV 2024] also a propose an approach that requires fine-tuning with a fast inference pass (compared to RePaint). Additionally, "Blended Latent Diffusion" [Avrahami et al., SIGGRAPH 2023] and "Uni-paint" [Yang et al., MM 2023] leverage latent diffusion models for an inpainting task (while the former does not explicitly target unconditional inapinting, it ca be used for that purpose). It would be very helpful to see comparisons to some of these recent works as well as a high-level discussion of their differences in order to evaluate the proposed method.
>
> We have included results from Blended Latent Diffusion (BLD) and Uni-paint (LatentPaint's implementation was not publicly available).
>
> Our analysis demonstrates that latent diffusion models struggle with inpainting tasks when masks are highly fine-grained. This issue arises due to the loss of critical information about the conditional and generative regions during the mask downsampling process. This impact of this limitation is most evident in the *Super-Res.2x* and *Alternating Lines* mask configurations, where baseline models consistently fail to produce satisfactory results.
>
> For *Wide* and *Narrow* masks, BLD demonstrates strong performance across all datasets, achieving KID metrics comparable to TD-Paint.
>
> For the larger masks (*Half* and *Expand*), both BLD and Unipaint fail to produce satisfactory results, as shown in Figures 4, 5, and 6. This failure arises from their methodology of combining noisy conditional masked-latent code with generated latent code. Similar to RePaint, this approach creates ambiguity in distinguishing between conditional and generated components of the latent code.
>
> These findings suggest that TD-Paint's principles could be adapted to work with latent diffusion models, potentially enhancing the capabilities of existing latent diffusion-based inpainting approaches, particularly when handling large masks.
>
> > On a more minor note, the presentation and exposition could be improved. Figure 1, which illustrates the proposed approach in contrast to RePaint, is somewhat difficult to follow and includes notation that is not defined until much later. In contrast, the idea behind RePaint is explained multiple times in various sections throughout the paper and could be truncated.
>
> We acknowledge your concern regarding the clarity of Figure 1. We have redesigned the figure and caption to improve its readability, introducing all relevant notation directly in the caption.
>
> > I would also encourage the authors to cite and discuss "Differential Diffusion: Giving Each Pixel Its Strength" [Levin and Fried, 2023]. Given that this work does not appear in a peer-reviewed publication, I am not considering it in my rating, but given its close relevance to the topic, it would be good to mention.
>
> We thank the reviewer for bringing this paper to our attention. We have added a discussion of this work in the Related Work section.
> We think that the fundamental distinction between Differential Diffusion and TD-Paint lies in their approach to noise application during the diffusion process.
> While Differential Diffusion maintains uniform noise levels across all regions of the input during each step, TD-Paint implements varying noise levels, enabling the usage of conditional information from the initial stages.
> Specifically, Differential Diffusion cannot leverage conditional information during initial steps, whereas TD-Paint's allows for immediate incorporation of conditional knowledge.
> Future research could explore the potential combinations of TD-Paint and complementary approaches such as Differential Diffusion, potentially leading to more robust and flexible diffusion models.

---

> > ### Comment · Reviewer_XQUX · 2024-11-26
> > **Response**
> >
> > Thank you to the authors for their response. Most of my concerns have been addressed, so I have raised my score.

---

### Author Response · Authors · 2024-11-24
**Thank you for your reviews**

We thank the reviewers for their constructive feedback that can improve our manuscript. We are pleased that the reviewers recognize the value of TD-Paint's core contribution (XQUX, 8fx5), acknowledging both its intuitive nature (XQUX) and effectiveness (XQUX, 8fx5, DSku, VMkS, DEmo). We particularly appreciate the recognition of our manuscript's clarity and comprehensive experimental validation (8fx5, DSku, VMkS).

We have carefully addressed all reviewers' comments and concerns through additional experiments, expanded baseline comparisons, enhanced discussion of related works, and improved writing clarity. Specifically, we have made the following changes in the updated pdf:
- Figure 1 has been revised to clarify the notations and expand the caption, providing better context and improving comprehension (addressing Reviewer XQUX)
- Extended Related Work section with a detailed discussion on Differential Diffusion (addressing Reviewer XQUX)
- Relocated the discussion of CNN- and transformer-based inpainting models to the appendix to improve the flow
- Added comprehensive comparisons with recent state-of-the-art methods:
  * Latent Diffusion Inpainting [1]
  * ControlNet-Inpaint [2]
  * Blended Latent Diffusion [3]
  * Uni-Paint [4]
  * PowerPaint [5]
  (addressing Reviewers XQUX, 8fx5, DSku)
- Incorporated corresponding discussions in the experimental results section

[1] "High-resolution image synthesis with latent diffusion models." CVPR 2022

[2] "Adding conditional control to text-to-image diffusion models." ICCV 2023

[3] "Blended latent diffusion." ACM TOG 2023

[4] "Uni-paint: A unified framework for multimodal image inpainting with pretrained diffusion model." ACM International Conference on Multimedia. 2023

[5] "A task is worth one word: Learning with task prompts for high-quality versatile image inpainting." ECCV 2025

---

### Meta-Review · Area_Chair_RZr7 · 2024-12-23

**Metareview:**

The paper introduces TD-Paint, a novel approach for faster diffusion-based inpainting by leveraging time-aware pixel conditioning. It introduces a spatially varying timestep map that allows the model to maintain known pixel regions while efficiently diffusing unknown areas, eliminating the need for repeated sampling as in RePaint. TD-Paint achieves significant speed improvements without architectural changes, maintaining or exceeding state-of-the-art performance across datasets. The approach is simple, effective, and demonstrated through extensive experiments.

Strengths:
Innovative Method: Introduces spatially varying timesteps for precise, efficient inpainting without architectural changes.
Faster Performance: Speeds up inpainting significantly compared to existing methods like RePaint.
Robust Results: Achieves high-quality inpainting across diverse datasets, masks, and scenarios.
Simple and Practical: Intuitive, easy-to-implement approach with broad applicability.

Weaknesses:
Limited Comparisons: Missing evaluations against recent state-of-the-art diffusion-based methods like LatentPaint and PowerPaint.
Generalization Issues: Trained on limited datasets, with unclear applicability to open-world scenarios or broader editing tasks.
Mask Handling: Struggles with precise masks and lacks evaluation on realistic, user-oriented masks.
Modest Gains: Visual improvements over baselines are minor, raising concerns about the method’s impact.

We are happy to accept the paper, as all reviewers gave positive scores.

**Additional Comments On Reviewer Discussion:**

The authors thoroughly addressed reviewer concerns with additional experiments, expanded baseline comparisons, improved discussion of related works, and clearer writing.

Score Increases (from Negative to Positive!):
Reviewer XQUX raised their score to positive, acknowledging that most concerns were addressed.
Reviewer DSku also raised their score, appreciating the detailed explanations and suggesting the authors refine claims about the pixel-wise map for clarity.

The revisions effectively resolved key issues, leading to improved reviewer satisfaction and supporting the decision to accept.

---

### Decision · Program_Chairs · 2025-01-22

Accept (Poster)